# In-Context Reinforcement Learning From Suboptimal Historical Data

**Juncheng Dong** [1]  **Moyang Guo** [1]  **Ethan X. Fang** [2]  **Zhuoran Yang** [3]  **Vahid Tarokh** [1]

## Abstract

Transformer models have achieved remarkable empirical successes, largely due to their in-context learning capabilities. Inspired by this, we explore training an autoregressive transformer for in-context reinforcement learning (ICRL). In this setting, we initially train a transformer on an offline dataset consisting of trajectories collected from various RL tasks, and then fix and use this transformer to create an action policy for new RL tasks. Notably, we consider the setting where the offline dataset contains trajectories sampled from suboptimal behavioral policies. In this case, standard autoregressive training corresponds to imitation learning and results in suboptimal performance. To address this, we propose the *Decision Importance Transformer* (DIT) framework, which emulates the actor-critic algorithm in an in-context manner. In particular, we first train a transformer-based value function that estimates the advantage functions of the behavior policies that collected the suboptimal trajectories. Then we train a transformer-based policy via a weighted maximum likelihood estimation loss, where the weights are constructed based on the trained value function to steer the suboptimal policies to the optimal ones. We conduct extensive experiments to test the performance of DIT on both bandit and Markov Decision Process problems. Our results show that DIT achieves superior performance, particularly when the offline dataset contains suboptimal historical data.

## 1. Introduction

Transformer models (TMs) have achieved remarkable empirical successes (Radford et al., 2019; OpenAI, 2024). In particular, TMs trained on vast amount of data have shown remarkable in-context learning (ICL) capabilities, solving new supervised learning tasks only with a few demonstrations and without requiring any parameter updates (Brown et al., 2020a; Akyürek et al., 2022). Meanwhile, substantial evidence demonstrates that autoregressive TMs excel at solving individual reinforcement learning (RL) tasks, where a TM-based policy is trained and tested on the *same* RL task (Li et al., 2023b). Inspired by these, recent research has explored the use of TMs for in-context RL (ICRL). In this setting, we pretrain TMs on an offline dataset consisting of trajectories collected from a family of different RL tasks. After pretraining, we deploy the pretrained TMs to solve *new and unseen* RL tasks (Laskin et al., 2022; Lee et al., 2024). See Figure 1 (a) and (c) for comparisons between standard offline RL and ICRL. When presented with a *context dataset* containing environment interactions collected by unknown and often suboptimal policies, pretrained TMs predict the optimal actions for current states from the environmental information within the context dataset. See Figure 1 for a visual illustration. Two recent works, *Algorithm Distillation* (AD) (Laskin et al., 2022) and *Decision Pretrained Transformer* (DPT) (Lee et al., 2024), have demonstrated impressive ICRL abilities, inferring near-optimal policies for new RL tasks.

**Challenges.** However, existing supervised pretraining approaches focus on training TMs to imitate the actions in the pretraining datasets and thus have *stringent requirements* on the pretraining datasets. For example, AD requires the pretraining dataset to capture the learning process of RL algorithms—from episodes generated by randomly initialized policies to those collected by nearly optimal policies—across a wide range of RL tasks; DPT requires access to optimal policies to generate a set of optimal action labels for its supervised pretraining of TMs. To overcome these limitations, this work considers pretraining TMs for ICRL *using only suboptimal historical data*. While this presents significant challenges, it also offers substantial potential benefits by significantly improving the feasibility of ICRL, as suboptimal trajectories are far easier to gather. For instance, large companies often maintain extensive databases

---

[1]Department of Electrical and Computer Engineering, Duke University, Durham, US [2]Department of Biostatistics and Bioinformatics, Duke University, Durham, US [3]Department of Statistics and Data Science, Yale University, New Haven, US. Correspondence to: Juncheng Dong <juncheng.dong@duke.edu>.

*Proceedings of the 42nd International Conference on Machine Learning*, Vancouver, Canada. PMLR 267, 2025. Copyright 2025 by the author(s).

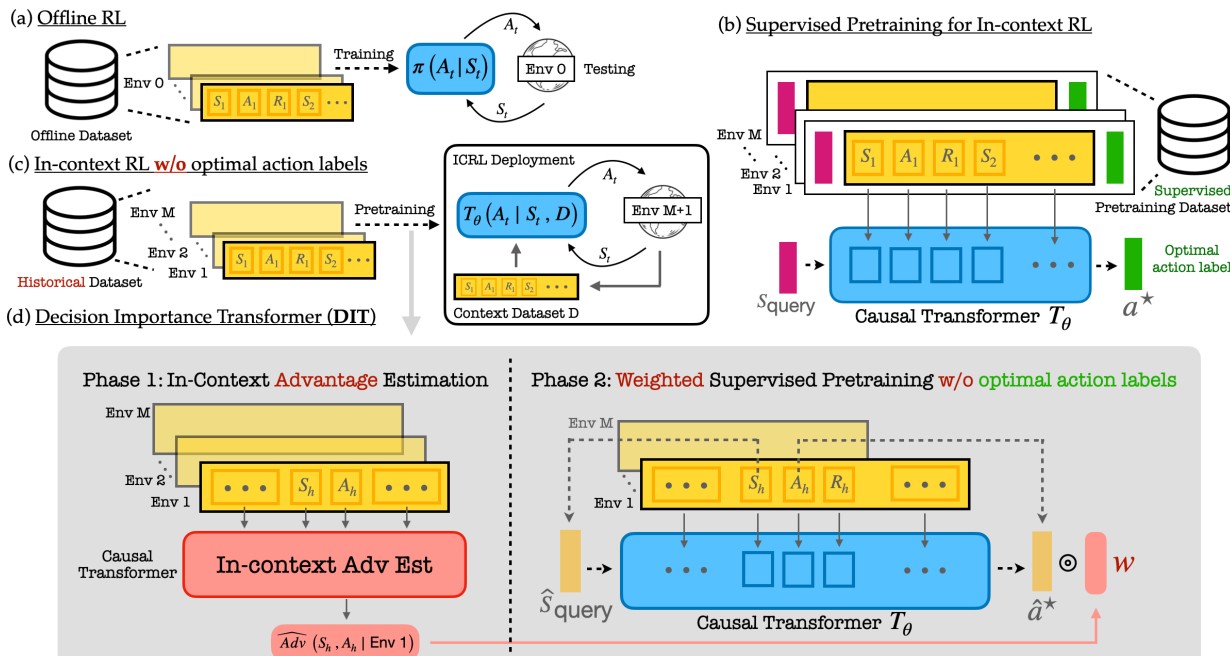

Figure 1: **(a) and (c) Comparison between Offline RL and ICRL.** Standard offline RL trains and tests a policy $\pi$ in the same task (*Env 0*); ICRL pretrains TMs on trajectories collected from a family of different RL tasks (*Env 1, Env 2, . . . , Env M*), and deploys the pretrained TMs to unseen tasks (*Env M+1*). **ICRL Deployment.** The pretrained TMs generate actions conditioned on the current states and *context datasets* consisting of offline trajectories collected by (suboptimal) behavioral policies from the unseen tasks. **(b) Supervised Pretraining.** Presented with offline trajectories and *optimal action labels*, TMs are pretrained to predict the optimal actions for query states across RL tasks. **(c) ICRL from Suboptimal Historical Data.** This work addresses the challenging problem of ICRL without optimal action labels. **(d) Schematic Overview of the Proposed Framework DIT.** Lack of the optimal action labels, the proposed framework employs *in-trajectory* state-action pairs as query states and *pseudo*-optimal action labels, and a *weighted* pretraining objective, where the weights are based on the optimality of actions, estimated by a TM-based in-context advantage function estimator.

of historical trajectories from non-expert users.

**Contributions.** In pursuit of this goal, we introduce *Decision Importance Transformer* (**DIT**), a supervised pretraining framework for ICRL using only *historical* trajectories collected by *suboptimal* behavioral policies across distinct RL tasks. When the pretraining datasets contain only suboptimal trajectories, existing approaches correspond to imitation learning and thus result in suboptimal performance. DIT overcomes this challenge through several techniques:

- DIT learns to infer near-optimal actions from suboptimal trajectories through an exponential reweighting technique that assigns *good actions* in the offline dataset with *more weights* during supervised pretraining. These assigned weights guide the suboptimal policies toward the optimal ones.

- In particular, the assigned weights are constructed from the advantage functions of the behavior policies such

that actions with high advantage values receive more weights during pretraining, leading to *guaranteed policy improvements* over the behavior policies.

- Notably, although advantage weighted regression has been studied in standard RL (Wang et al., 2018; Peng et al., 2019), it remains unclear how to generalize this approach to ICRL. The primary challenge is that the weighting function in ICRL must be *task-dependent*, thus requiring the estimation of advantage functions for *all* RL tasks in the pretraining dataset. To this end, the most significant technical difficulty arises from the unknown source tasks of pretraining trajectories, preventing us from grouping trajectories from the same RL tasks to improve estimation. As a result, we must estimate the advantage functions *individually* for each trajectory in the pretraining dataset.

- To address this formidable challenge, DIT trains a TM-based advantage estimator that interpolates across

trajectories from different tasks for an *in-context estimation* of the advantage functions to facilitate the weighted supervised pretraining framework. See Figure 1(d) for a visualization.

**Empirical Results.** Through extensive experiments on various bandit and Markov Decision Process (MDP) problems, we demonstrate that pretrained DIT models generalize to unseen decision-making problems. On bandit problems, the performance of DIT models matches that of the theoretically optimal bandit algorithms (e.g., Thompson Sampling (Russo et al., 2018)). In four challenging MDP problems including two navigating tasks with sparse rewards and two complex continuous control tasks, DIT models achieve superior performance, particularly when the pretraining dataset contains suboptimal trajectories. Notably, in many scenarios, DIT is comparable to DPT in both online and offline testings, despite being pretrained without optimal action labels.

## 2. Related Work

**Offline Reinforcement Learning.** Since we consider pretraining with historical data, our work falls within the broader field of offline RL. While online RL algorithms (Kaelbling et al., 1996; François-Lavet et al., 2018) learn optimal policies by interacting with the environments through trial and error, offline RL (Levine et al., 2020; Matsushima et al., 2020; Prudencio et al., 2023) aims to infer optimal policies from historical data collected by (sub-optimal) behavioral policies. One of the most substantial challenges for offline RL is the distribution shift caused by the mismatch between behavioral policies and optimal policies (Levine et al., 2020; Kostrikov et al., 2021). To this end, offline RL algorithms learn pessimistically by either policy regularization or underestimating the policy returns (Wu et al., 2019; Kidambi et al., 2020; Kumar et al., 2020; Rashidinejad et al., 2021; Yin & Wang, 2021; Jin et al., 2021; Fujimoto & Gu, 2021; Dong et al., 2023). While the goal of offline RL is to solve the *same* RL tasks from where the offline datasets are collected, the goal of ICRL is to efficiently generalize to *unseen* tasks after pretraining with offline datasets from diverse RL tasks.

**Transformer Models and Autoregressive Decision Making.** Large language models and autoregressive models (Radford et al., 2019; Brown et al., 2020b; Wu et al., 2023b; Touvron et al., 2023; OpenAI, 2024) have achieved astonishing empirical successes in a wide range of application areas, including medicine (Singhal et al., 2023; Thirunavukarasu et al., 2023), education (Kasneci et al., 2023), finance (Wu et al., 2023a; Yang et al., 2023), etc. As it is natural to use autoregressive models for sequential decision-making, transformer models have demonstrated superior performance in both bandit and MDP problems (Li et al., 2023a; Yuan et al., 2023). In particular, Decision

Transformer (DT) (Chen et al., 2021; Zheng et al., 2022; Liu et al., 2023; Yamagata et al., 2023) uses return-conditioned supervised learning to tackle offline RL. Although salable to multi-task settings (i.e., one model for multiple RL problems), DT is commonly criticised for its inability to improve upon the offline datasets and provably sub-optimal in certain scenarios, e.g., environment with high stochasticity (Brandfonbrener et al., 2022; Yang et al., 2022; Yamagata et al., 2023). To this end, AD (Laskin et al., 2022) uses sequential modeling to emulate the learning process of RL algorithms, i.e., meta-learning (Vilalta & Drissi, 2002). The work most closely related to ours is DPT, a supervised pretraining approach for in-context decision making (Lee et al., 2024). DPT trains transformers to predict the optimal action given a query state and a set of transitions. Both AD and DPT have stringent assumptions on the pretraining datasets. Our work overcomes those drawbacks and does not require query to optimal policies nor the learning histories of RL algorithms (Laskin et al., 2022; Lee et al., 2024).

## 3. Preliminary

**Markov Decision Process.** Sequential decision problems can be formulated as Markov Decision Processes (MDPs). An MDP $\tau$ is described by the tuple $(\mathcal{S}, \mathcal{A}, P_\tau, R_\tau, \gamma, \rho_\tau)$ where $\mathcal{S}$ is the set of all possible states, $\mathcal{A}$ is the set of all possible actions, $P_\tau : \mathcal{S} \times \mathcal{A} \to \Delta(\mathcal{S})$ is the transition function that describes the distribution of the next state, $R_\tau : \mathcal{S} \times \mathcal{A} \to \mathbb{R}$ is the reward function, $\gamma \in (0, 1)$ is the discounting factor for cumulative rewards, and $\rho_\tau \in \Delta(\mathcal{S})$ is the initial state distribution. An agent interacts with the environment $\tau$ as follows. At the initial step $h = 1$, an initial state $s_1 \in \mathcal{S}$ is sampled according to $\rho_\tau$. At each time step $h$, the agent chooses action $a_h \in \mathcal{A}$ and receives reward $r_h = R_\tau(s_h, a_h)$. Then the next state $s_{h+1}$ is generated following $P_\tau(s_h, a_h)$. A policy $\pi : \mathcal{S} \to \Delta(\mathcal{A})$ maps the current state to an action distribution. Let $G_\tau(\pi) = \mathbb{E}[\sum_{h=1}^{\infty} \gamma^{h-1} r_h | \pi, \tau]$ denote the expected cumulative reward of $\pi$ for task $\tau$. The goal of an agent is to learn the optimal policy $\pi_\tau^\star$ that maximizes $G_\tau(\pi)$.

**Decision-Pretrained Transformer.** Our proposed approach builds upon the model architecture of DPT, which is a supervised pretraining method for TMs to have ICRL capabilities (see Figure 1(b) for its architecture). DPT assumes a set of tasks $\{\tau^i\}_{i=1}^m$ sampled independently from a task distribution $p_\tau$, with each $\tau^i$ as an instance of MDP. For each task $\tau^i$, a context dataset $D^i$ is sampled, consisting of interactions between a behavioral policy and $\tau^i$. That is, $D^i = \{(s_h^i, a_h^i, s_{h+1}^i, r_h^i)\}_h$, where $a_h^i$ is chosen by a behavioral policy. Additionally, for each task $\tau^i$, a query state $s_{\text{query}}^i \in \mathcal{S}$ is sampled, and an associated optimal action label $a_i^\star$ is sampled from $\pi_{\tau^i}^\star(s_{\text{query}})$, where $\pi_{\tau^i}^\star$ is the optimal policy for $\tau^i$. The complete pretraining dataset is

$\mathcal{D}_{pre} = \{D^i, s^i_{\text{query}}, a^\star_i\}^m_{i=1}$. Let $T_\theta$ denote a causal transformer with parameters $\theta$ (Radford et al., 2019). The pretraining objective of DPT is defined as

$$\min_\theta \frac{1}{m} \sum_{i=1}^m -\log T_\theta\left(a^\star_i | s^i_{\text{query}}, D^i\right). \tag{1}$$

**ICRL Deployment.** After pretraining, the pretrained autoregressive TM $T_\theta$ can be deployed as both an online and offline agent. During deployment, an unseen testing task $\tau$ is sampled from $p_\tau$. For offline deployment, a dataset $D_{\text{off}}$ is first sampled from $\tau$, e.g., $D_{\text{off}}$ contains trajectories gathered from a behavioral policy in $\tau$, then DPT follows the policy $T_\theta(\cdot|s_h, D_{\text{off}})$ after observing the state $s_h$ at time step $h$. For online deployment, DPT initiates with an empty dataset $D_{\text{on}}$. In each episode, DPT follows the policy $T_\theta(\cdot|s_h, D_{\text{on}})$ to collect a trajectory $\{s_1, a_1, r_1, \ldots, s_H, a_H, r_H\}$ which will be appended into $D_{\text{on}}$. This process repeats for a predefined number of episodes. See Algorithm 2 in appendix for the pseudocodes of both offline and online deployments.

## 4. Decision Importance Transformer

Here we introduce our proposed framework *Decision Importance Transformer* (**DIT**).

**Pretraining with Suboptimal Data.** Similar to DPT, DIT assumes a family of datasets $\mathcal{D} = \{D^i\}^m_{i=1}$ where $D^i$ consists of $H$ transitions $\{(s^i_h, a^i_h, s^i_{h+1}, r^i_h)\}^H_{h=1}$ collected by the (suboptimal) behavioral policy $\pi^b_{\tau^i}$ in task $\tau^i$ which itself is independently sampled from the task distribution $p_\tau$. In contrast to DPT, however, DIT *does not require* the set of paired query states and optimal action labels $\{s^i_{\text{query}}, a^\star_i\}^m_{i=1}$, which are often difficult to obtain in practice.

**Notations.** In the sequel, for any task $\tau$, we assume that it has an index (parameter) also denoted by $\tau$ such that the task information $\tau$ can be an explicit input to a meta-policy $\pi(s|a; \tau)$ which can generate distinct policies based on the received task $\tau$. For example, in robotic control tasks, $\tau$ may represent the physical parameters of the robots such as robot mass or the environmental parameters such as ground friction. We use $\pi^b_\tau(a|s)$ to denote the *behavioral policy* for task $\tau$. Denote

$$V^b_\tau(s) = \mathbb{E}\left[\sum_{h=1}^\infty \gamma^{h-1} r_h \Big| s_1 = s, \tau, \pi^b_\tau\right],$$

$$Q^b_\tau(s, a) = \mathbb{E}\left[\sum_{h=1}^\infty \gamma^{h-1} r_h \Big| s_1 = s, a_1 = a, \tau, \pi^b_\tau\right]$$

as its value and action-value functions respectively, and let $A^b_\tau(s, a) = Q^b_\tau(s, a) - V^b_\tau(s)$ be its *advantage function*.

For presentation clarity, in Section 4.1, we first consider the scenarios where **(i)** $A^b_\tau(s, a)$ is known and **(ii)** the task

index $\tau$ is also known and can be provided as input to a meta-policy. Then in Section 4.2, we introduce solutions for scenarios where $A^b_\tau(s, a)$ and $\tau$ need to be estimated. All proofs of the theoretical results in this section are deferred to Appendix C.

### 4.1. Weighted Maximum Likelihood Estimation

**Motivation.** To motivate DIT, we first consider the setting of imitation learning where the agent is trained and tested on the same task. Given a dataset of transitions $D = \{(s_h, a_h, s_{h+1}, r_h)\}$ collected by a behavior policy $\pi^b(a|s)$ with advantage function $A^b(s, a)$, Wang et al. (2018) proposes to optimize a weighted objective:

$$\arg\max_\pi \sum_{(s_h, a_h) \in D} \exp(A^b(s_h, a_h)) \cdot \log \pi(a_h|s_h).$$

The rationale is that the good actions in the offline dataset, that is, $a_h$ with high advantage value $A^b(s_h, a_h)$, should be given more weights during imitation learning. These weights essentially work as importance sampling ratios so that the action distribution is closer to the optimal one.

**Weighted Pretraining for ICRL.** In contrast to imitation learning that focuses on individual RL tasks, the objective of DIT is to learn a task-conditioned policy $\pi(a|s; \tau)$ with the task index $\tau$ as input. In particular, $\pi(a|s; \tau)$ should perform well for $\tau \sim p_\tau$.

Motivated by the aforementioned weighted imitation learning objective, DIT has the following *weighted maximum likelihood estimation* (**WMLE**) loss for pretraining:

$$L(\pi) = -\mathbb{E}_{\tau, s, a}\left[\exp\left(\frac{A^b_\tau(s, a)}{\eta}\right) \log \pi(a|s; \tau)\right]. \tag{2}$$

The expectation in Equation (2) is with respect to $\tau \sim p_\tau$, $s \sim d_\tau(s)$, and $a \sim \pi^b_\tau(a|s)$ where $d_\tau(s)$ is the discounted visiting frequencies of $\pi^b_\tau(a|s)$ defined as $d_\tau(s) = (1-\gamma)\mathbb{E}\left[\sum_{h=1}^\infty \gamma^{h-1}\mathbb{1}\{s_h = s\}|\tau, \pi^b_\tau\right]$. The effectiveness of the objective in Equation (2) is demonstrated by the following result which states that the optimizer to DIT's pretraining objective is also the solution to another policy optimization problem that is easier to interpret.

**Proposition 4.1.** *Consider the following optimization problem where $\mathbb{E}_{\tau, s, a}$ is defined as in Equation (2) except that $a \sim \pi(a|s; \tau)$, i.e., the action is sampled from the task-conditioned policy rather than the behavioral policies:*

$$\max_\pi J(\pi) = \mathbb{E}_{\tau, s, a}\left[\underbrace{A^b_\tau(s, a)}_{(I)} - \eta \cdot \underbrace{D_{\text{KL}}(\pi(\cdot|s; \tau)\|\pi^b_\tau(\cdot|s))}_{(II)}\right],$$

(3)

*where $D_{\text{KL}}$ is the Kullback–Leibler (KL) divergence, and let $\pi^\star \in \arg\max_\pi J(\pi)$ be its optimizer. Then we have for*

*any policy $\pi(a|s;\tau)$,*

$$\mathbb{E}_{\tau \sim p(\tau), s \sim d_\tau(s)} \left[ D_{\mathrm{KL}} \left( \pi^\star(\cdot|s;\tau) \| \pi(\cdot|s;\tau) \right) \right]$$

$$= -\mathbb{E}_{\tau,s,a} \left[ \frac{1}{Z_\tau(s)} \exp \left( A_\tau^b(s,a)/\eta \right) \cdot \log \pi(a|s;\tau) \right] + C, \tag{4}$$

*where $\mathbb{E}_{\tau,s,a}$ is defined as in (2), $C$ is a constant independent of $\pi$ and $Z_\tau(s) = \sum_a \pi_\tau^b(a|s) \exp(A_\tau^b(s,a)/\eta)$.*

In Equation (3), the objective is to find a policy $\pi^\star$ that improves over the behavior policy (by maximizing term (I)) and does not stray too far from the behavior policy (by minimizing term (II)). When the behavioral policy $\pi_\tau^b(a|s)$ is near-optimal, $\eta$ should set to a large value so that we can have *safe* improvements over the behavioral policy. On the other hand, when the behavioral policy is highly sub-optimal, $\eta$ should set to a small value so that we have more freedom for policy improvement to decrease the sub-optimality. Note that the $D_{\mathrm{KL}}$ constraint (term (II) in Equation (3)) is critical for pretraining large transformer models to prevent policy collapse (Schulman, 2015).

Comparing Equation (4) with the pretraining objective of DIT in Equation (2), we observe that DIT aims to identify a policy that is closest to $\pi^\star$ by setting $Z_\tau(s) = 1$ (we provide a brief discussion for why this is valid in Appendix C.3). When $A_\tau^b(s,a)$ is known, the pretraining objective of DIT can be estimated with the given pretraining dataset $\mathcal{D}$ by minimizing the following loss function

$$L_n(\pi) := -\frac{1}{mH} \sum_{i=1}^{m} \sum_{h=1}^{H} w_h^i \log \pi \left( a_h^i | s_h^i; \tau^i \right), \tag{5}$$

where $w_h^i = \exp \left( A_{\tau^i}^b(s_h^i, a_h^i)/\eta \right)$. Next we establish that DIT can provably achieve policy improvement.

**Proposition 4.2** (Policy Improvement). *Let $\pi^\star$ be the policy that optimizes (3). For any task $\tau$ and policy $\pi$, let $G_\tau(\pi) = \mathbb{E}[\sum_{h=0}^{\infty} \gamma^h r_h | \pi, \tau]$ represent the expected cumulative reward of $\pi$ for $\tau$. Let $\pi_\tau^\star$ denote $\pi^\star(a|s;\tau)$. Then we have*

$$\mathbb{E}_{\tau \sim p_\tau} [G_\tau(\pi_\tau^\star) - G_\tau(\pi_\tau^b)] \geq \frac{\eta}{1-\gamma} \mathbb{E}_{\tau \sim p_\tau} [C_\tau^D]$$

$$- \frac{2\gamma}{(1-\gamma)^2} \mathbb{E}_{\tau \sim p_\tau} \left[ C_\tau^A \cdot \sqrt{C_\tau^D/2} \right], \tag{6}$$

*where $C_\tau^D = \mathbb{E}_{s \sim d_\tau(s)} [D_{\mathrm{KL}}(\pi^\star(\cdot|s;\tau) \| \pi_\tau^b(\cdot|s))]$ and $C_\tau^A = \max_s |\mathbb{E}_{a \sim \pi^\star(a|s;\tau)} A_\tau^b(s,a)|$.*

In particular, when the magnitude of the advantage function $A_\tau^b$ is small, the right-hand side of Equation (6) is nonnegative. In this case, the policy $\pi^\star$ obtained by solving Equation (3) is strictly better than the behavior policy. Equivalently, adding the exponential weights in Equation (5) is strictly better than vanilla imitation learning, when the total number of pretraining tasks $m$ is large.

## 4.2. In-context Task Identification and Advantage Function Estimation

However, two key challenges remain: **(i)** During *pretraining*, the advantage function $A_\tau^b(s,a)$ is not accessible for generating the weights required by the WMLE loss; **(ii)** During *deployment*, the task index $\tau$ is not accessible as only a context dataset $D_\tau$ is presented. In other words, the true identity of the testing task $\tau$ is unknown.

**In-context Task Identification.** To address the second problem, we follow DPT to instantiate $\pi(a|s;\tau)$ with an autoregressive transformer $T_\theta$ parameterized by $\theta$. Conditioned on a given context dataset $D_\tau$ consisting of environment interactions collected by a behavioral policy in $\tau$, the TM-based policy $T_\theta(a|s, D_\tau)$ first implicitly extracts task information about $\tau$ from the context $D_\tau$ and chooses an action based on the extracted task information (see Lee et al. (2024) for a detailed discussion). During pretraining, $T_\theta$ learns to extract useful task information for the pretraining tasks $\{\tau^i\}_{i=1}^m$ conditioned on the pretraining context datasets $\{D^i\}_{i=1}^m$, and generalizes to unseen tasks during testing.

**In-context Advantage Function Estimation.** The first problem is more critical. Given that during pretraining the context dataset $D^i$ may contain up to several trajectories for each task $\tau^i$ in the setting of ICRL, *estimation of $A_{\tau^i}^b(s,a)$ based on $D^i$ alone can be unreliable*. To this end, in the same spirit of ICRL, we propose to use an *in-context advantage function estimator* $\widehat{A}_b(s_h^i, a_h^i | \tau^i)$ to estimate the advantage value of any state-action pair $(s_h^i, a_h^i)$ in the pretraining dataset $\mathcal{D}$. Specifically, $\widehat{A}_b(s_h^i, a_h^i | \tau^i)$ is implemented by two TMs:

$$\widehat{A}_b(s_h^i, a_h^i | \tau^i) = \widehat{Q}_\zeta(s_h^i, a_h^i | D_Q^{i,h}) - \widehat{V}_\phi(s_h^i | D_V^{i,h}), \tag{7}$$

where $\widehat{V}_\phi$ and $\widehat{Q}_\zeta$ are two transformers, parameterized by $\phi$ and $\zeta$, acting as the in-context value and action value estimators that interpolate across tasks to have an improved estimation.

**Model Architecture.** Let $G_h^i = \sum_{h'=h}^{H} \gamma^{h'-h} r_h^i$ be the in-trajectory discounted cumulative reward starting from step $h$. For any observed state-action pair $(s_h^i, a_h^i)$ in the pretraining dataset, $\widehat{Q}_\zeta(s_h^i, a_h^i | D_Q^{i,h})$ and $\widehat{V}_\phi(s_h^i | D_V^{i,h})$ estimate the action-value function $Q_{\tau^i}^b(s_h^i, a_h^i)$ and value function $V_{\tau^i}^b(s_h^i)$ respectively, conditioned on the histories of transitions $D_Q^{i,h} = \{(s_j^i, a_j^i, G_j^i)\}_{j=1}^{h-1}$ and $D_V^{i,h} = \{(s_j^i, G_j^i)\}_{j=1}^{h-1}$, where we employ $\{G_j^i\}_{j<h}$ as the noisy labels for value functions to facilitate in-context learning. See Figure 6 for their visual representations.

**Training.** We train $\widehat{V}_\phi$ and $\widehat{Q}_\zeta$ with the following objective function:

$$\min_{\phi,\zeta} L_A(\phi,\zeta) := L_{\mathrm{reg}}(\phi,\zeta) + \lambda \cdot \left( L_V^B(\phi) + L_Q^B(\zeta) \right), \tag{8}$$

where $\lambda > 0$ is a hyperparameter to balance

$$L_{\text{reg}}(\phi, \zeta) := \frac{1}{mH} \sum_{i=1}^{m} \sum_{h=1}^{H} \left( \widehat{V}_\phi(s_h^i | D_V^{i,h}) - G_h^i \right)^2$$
$$+ \left( \widehat{Q}_\zeta(s_h^i, a_h^i | D_Q^{i,h}) - G_h^i \right)^2,$$

$$L_Q^B(\zeta) := \frac{1}{mH} \sum_{i,h} \left( \widehat{Q}_h^i(\zeta) - \widehat{Q}_\zeta(s_{h+1}^i, a_{h+1}^i | D_Q^{i,h+1}) \right)^2$$

where $\quad \widehat{Q}_h^i(\zeta) := r_h^i + \gamma \widehat{Q}_\zeta(s_h^i, a_h^i | D_Q^{i,h}), \quad$ and

$$L_V^B(\phi) := \frac{1}{mH} \sum_{i,h} \left( \widehat{V}_h^i(\phi) - \widehat{V}_\phi(s_{h+1}^i | D_V^{i,h+1}) \right)^2$$

where $\quad \widehat{V}_h^i(\phi) = r_h^i + \gamma \widehat{V}_\phi(s_h^i | D_V^{i,h}).$

Here, $L_Q^B$ and $L_V^B$ regularize the transformer models with the Bellman equations for value functions.

**DIT with In-context Advantage Estimator.** After training, with $\widehat{A}_b(s_h^i, a_h^i | \tau^i)$ defined in Equation (7) as an estimation of the true advantage function, we can now optimize the objective function of DIT to have the pretrained TM $T_{\theta^\star}$ for ICRL, i.e.,

$$\theta^\star \in \arg\min_{\theta \in \Theta} -\frac{1}{mH} \sum_{i,h} w_h^i \log T_\theta \left( a_h^i | s_h^i, D^i \right), \quad (9)$$

where $w_h^i = \exp(\widehat{A}_b(s_h^i, a_h^i | \tau^i)/\eta)$. We summarize the complete procedure of DIT in Algorithm 1.

# 5. Experiments

We empirically demonstrate the efficacy of DIT through experiments on various bandit and MDP problems. In bandit problems, DIT showcases matching performance to that of the theoretically optimal bandit algorithms in both online and offline settings. In MDP problems, we corroborate that DIT can infer close-to-optimal policies from suboptimal pretraining datasets. Notably, albeit without optimal action labels during pretraining, DIT models demonstrate performance as strong as that of DPT, which has access to optimal action labels during pretraining.

**Implementation.** We follow Lee et al. (2024) to choose GPT-2 (Radford et al., 2019) as the backbone for $T_\theta$, $\widehat{Q}_\xi$, and $\widehat{V}_\phi$ due to limited computation resource, and note that the performance may be further improved with larger models. We set $\gamma = 0.8$ for all tasks. We choose $\eta = 1$ for all tasks. Due to space constraint, see Appendix G for more details.

## 5.1. Bandit Problems

We consider linear bandit (LB) problems with an underlying structure shared among tasks. Specifically, there exists a bandit feature function $\phi : \mathcal{A} \to \mathbb{R}^d$ that is *fixed*

across tasks where $d$ denotes the dimension of linear bandit problems. The reward of a bandit $a \in \mathcal{A}$ in task $\tau^i$ is $r^i(a) \sim \mathcal{N}\left(\mu_a^i, \sigma^2\right)$ where $\mu_a^i = \mathbb{E}[r|a, \tau^i] = \langle \theta^i, \phi(a) \rangle$ and $\sigma^2 = 0.3$. Here, $\theta^i$ is the task-specific parameter that defines task $\tau^i$. We conduct experiments on LB problems where $K = 20$, $d = 10$ and $H = 200$. The pretraining dataset for DIT are generated as follows.

**Pretraining Dataset.** For LB problems, we generate the feature function $\phi : \mathcal{A} \to \mathbb{R}^d$ by sampling bandit features from independent Gaussian distributions, i.e., $\phi(a) \sim \mathcal{N}_d\left(0, I_d/d\right)$ for all $a \in \mathcal{A}$. To generate the pretraining tasks $\{\tau^i\}$, we sample their parameters $\{\theta^i\}$ independently following $\theta^i \sim \mathcal{N}_d\left(0, I_d/d\right)$. To generate context dataset $D^i$, we randomly generate a behavioral policy by mixing (i) a probability distribution samples a Dirichlet distribution and (ii) a point-mass distribution on one random arm. The mixing weights are uniform sampled from $\{0.0, 0.1, \ldots, 1.0\}$. At every time step $h$, the behavioral policy samples an action $a_h^i$ and receives $r_h^i$. *We do not enforce extra coverage of the optimal actions for bandit problems*. Following the setting of DPT (Lee et al., 2024), we collect 100k context datasets for LB problems.

**Comparisons.** We compare to the following baselines (see Appendix B for more details): ***Empirical Mean (EMP)*** selects the bandit with the highest average reward; ***Upper Confidence Bound (UCB)*** (Auer, 2002) builds upper confidence bounds for all bandits and selects the bandit with the highest upper bound; ***Lower Confidence Bound (LCB)*** (Xiao et al., 2021) builds lower confidence bounds for all bandits and selects the bandit with the highest lower bound; ***Thompson Sampling (TS)*** (Russo et al., 2018) builds a posterior distribution for the rewards of all bandits and selects the bandit with the highest sampled mean. In terms of metrics, for offline learning, we follow the convention to use the *suboptimality* defined as $(\mu_{a^\star} - \mu_{\hat{a}})$ where $\mu_{a^\star}$ is the mean reward of the optimal bandit and $\mu_{\hat{a}}$ is the mean reward of the chosen bandit; for online learning we use the *cumulative regret* defined as $\sum_h (\mu_{a^\star} - \mu_{a_h})$ where $a_h$ is the chosen action at time step $h$.

**Empirical Results.** In Figure 2, DIT models demonstrate superior performance in the online setting to those of the theoretically optimal bandit algorithms, i.e., UCB and TS. Deployed for unseen bandit problems, DIT models quickly identify the optimal bandits at the beginning and maintain low regrets over the horizon. In the offline setting, DIT can infer near-optimal bandits from trajectories collected by suboptimal policies. When the behavioral policies (captioned as *BEH*) are randomly generated policies, *DIT significantly outperforms both TS and LCB*, the theoretically optimal algorithm for bandit problems. When the context is collected by expert policies, *DIT models further improve upon their performance*. We also observe that DIT is slightly outper-

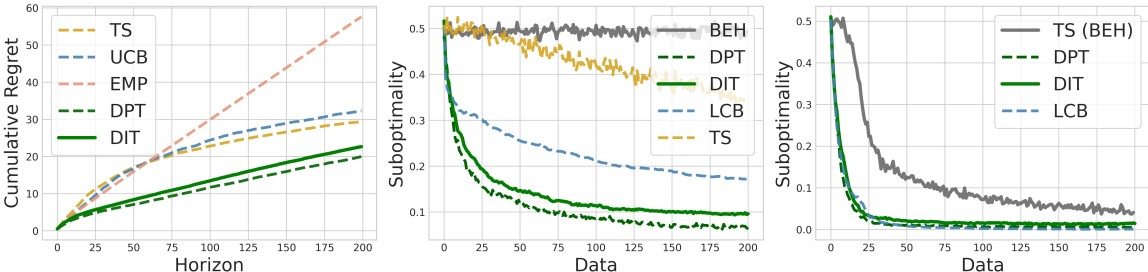

Figure 2: Results for Linear Bandits (lower values indicate better performance). **Left**: Online testing. **Middle**: Offline testing conditioned on trajectories gathered by highly suboptimal, randomly generated policies. **Right:** Offline testing condtioned on trajectories gathered by experts.

formed by DPT. This is expected given that DPT uses the optimal bandit information during pretraining. However, the loss curves of DPT and DIT demonstrate similar trend, showcasing the effectiveness of DIT's weighted pretraining.

## 5.2. MDP Problems

**Environments.** We conduct experiments on four challenging MDP environments: two navigating tasks with sparse reward Dark Room (Laskin et al., 2022) and Miniworld (Chevalier-Boisvert et al., 2023), as well as two complex continuous-control tasks Meta-world (reach-v2) (Yu et al., 2020) and Half-Cheetah (velocity) (Todorov et al., 2012). In **Dark Room**, the agent is randomly placed in a room of $10 \times 10$ grids with an *unknown* goal location on one of the grid. The agent needs to move to the goal location by choosing from 5 actions in 100 steps. In **Miniworld**, the agent is placed in a room and receives a $(25 \times 25 \times 3)$ color image and its direction as input. It can choose from four possible actions to reach a target box, out of four boxes of different colors. In **Meta-World**, the task is to control a robot hand to reach a target position in 3D space. In **Half-Cheetah**, the agent controls a robot to reach a target velocity, which is uniformly sampled from the interval $[0, 3]$, and is penalized based on how far its current velocity is from the target velocity. See Appendix D for more details.

**Pretraining Datasets**. For Dark Room and Miniworld, to ensure coverage of optimal actions (so that optimal policies can be inferred), at every step, with probability $p$ (respectively $1 - p$) we use optimal policy (respectively random policy) to choose action. We choose $p$ so that the average reward of the trajectories in the pretraining dataset is less than $30\%$ of that of the optimal trajectories. For Meta-World and Half-Cheetah, we construct the pretraining datasets using historical trajectories generated by *Soft Actor Critic* (SAC). Specifically, SAC is trained until convergence for each task, then we sample from its learning trajectories to build the dataset. Our SAC model training follows the settings outlined in Haarnoja et al. (2018). See Appendix E for details.

**Comparisons.** We compare **DIT** to other in-context algorithms as well as RL algorithms without pretraining. The baseline algorithms are briefly described next (see their implementation details in Appendix B).

- **Soft Actor Critic (SAC)** (Haarnoja et al., 2018): SAC is an online RL algorithm that trains an agent from scratch in every environment.

- **Algorithm Distillation (AD)**: AD is a sequence modeling-based approach for ICRL that emulates the learning process of RL algorithms (Laskin et al., 2022). To this end, AD requires the pretraining dataset to consist of complete learning histories of an RL algorithm —from episodes generated by randomly initialized policies to those collected by nearly optimal policies— across a wide range of RL tasks. In this work we use SAC as the RL algorithm for AD to emulate.

- **Decision Pretrained Transformer (DPT)**: DPT and DIT use the *same* context datasets for pretraining[1]. However, DPT requires query states and their associated optimal action labels across different tasks.

- **Prompt-DT (PDT)**: PDT is a Decision Transformer-based approach (Xu et al., 2022), which leverages the transformer's prompt framework for few-shot adaptation. PDT uses the same pretraining dataset as DIT. Thus, *the performance gain of DIT over PDT highlights the effectiveness of DIT's design*.

- **Behavior Cloning (BC)**: We include an variation of DIT without the exponential reweighting. This approach closely imitates BC, with the following pre-

---

[1]Note that the context datasets in our experiments are collected using suboptimal policies, as opposed to the uniformly random policies used by DPT in Lee et al. (2024). As a result, the reported performance of DPT in the Dark Room environment differs from that in the original paper.

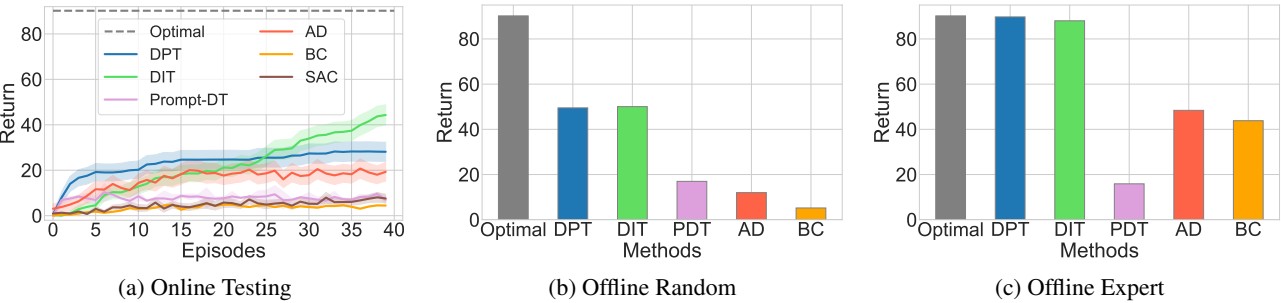

Figure 3: Results on **Dark Room** (higher values indicate better performance). **(a)**: Change in return of policies with additional online episodes for (in-context) learning. **(b)** and **(c)**: Offline evaluations with context trajectories sampled from random and expert policies.

training objective:

$$\min_{\theta} \frac{1}{mH} \sum_{i=1}^{m} \sum_{h=1}^{H} - \log T_{\theta} \left( a_h^i | s_h^i, D^i \right).$$

In particular, **AD** and **DPT** require *extra information* during pretraining: AD requires the complete learning history of RL algorithms while DPT requires optimal action labels. Given that DIT only relies on suboptimal historical data, the comparison is *inherently unfair*. Notably, despite these disadvantages, DIT outperforms AD and matches with DPT in most scenarios. In terms of metrics, we follow the convention to use the *episode cumulative return* $\sum_{h=1}^{H} r_h$.

**In-context Decision-making for Navigating Tasks.** We explore how our method generalizes to unseen RL tasks, using the Dark Room environment (Laskin et al., 2022). Following the evaluation protocol of DPT (Lee et al., 2024), we use 80 goals for training and evaluate on the remaining 20 unseen goals. For SAC, since it is an online learning method, we directly train from scratch on the 20 goals to benchmark the returns of ICRL. Figure 3a shows the online evaluation over 40 episodes. After 40 episodes, SAC gains little in return, demonstrating the difficulty of the RL tasks for testing. Restricted by their capability to efficiently explore in new tasks, BC also perform poorly. *Although our method (DIT) initially has lower returns than DPT and AD, it quickly surpasses them and continues to improve.* Figures 3b and 3c show the results for offline evaluations with expert (high-reward) trajectories and random (low-reward) trajectories. Despite being pretrained without the optimal action labels, DIT models demonstrate competitive performance to that of DPT.

**In-context Continuous Control.** We explore two complex continuous control tasks, Meta-World (Yu et al., 2020) and Half-Cheetah (Todorov et al., 2012). Meta-World has 20 tasks in total, to evaluate our approach's ability to generate to new RL tasks, we use 15 tasks to train and 5 to test. Sim-

ilarly, for Half-Cheetah, out of the 40 total tasks, we use 35 tasks to train and 5 to test. Due to space constraints, the results for Meta-World and Half-Cheetah are presented in Figure 5 in the Appendix. We observe that DIT outperforms PDT and BC in all testing scenarios. Moreover, DIT consistently outperforms AD despite with less information used for pretraining. It can also be observed that the performance gap between DPT and DIT is larger in the Meta-World environment compared to Half-Cheetah. We believe this is because Meta-World is a more challenging environment than Half-Cheetah. As a result, the *additional* set of optimal action labels for out-of-trajectory query states used by DPT has a greater impact on performance, while DIT can only utilize in-trajectory states and actions as query states with pseudo-optimal labels.

**Ablation Study on Weighted Supervised Pretraining.** While DIT's significantly improved performance over BC (the unweighted version of DIT) already demonstrates the effectiveness of the proposed weighted pretraining objective, we now conduct experiments in the Miniworld (Chevalier-Boisvert et al., 2023) environment to explore whether DIT reaches the *limits* of the weighted pretraining framework.

To this end, we compare our model to the DPT model that uses a pretraining dataset containing only query states that belong to the set of observed states in the pretraining dataset, along with their associated optimal action labels. In this scenario, the total number of pretraining context datasets and optimal action labels for DPT remains the same, but the query states are restricted. This restriction makes the DPT model function as an *oracle upper bound* for DIT, as all query states used by DIT in the weighted pretraining originate from the observed states.

The significant performance gain of DIT over BC (the unweighted version of DIT) demonstrate the effectiveness of the weighted pretraining framework. Surprisingly, in the online setting, DPT struggles to perform, while DIT models gradually improve their returns, as shown in Figure 4.

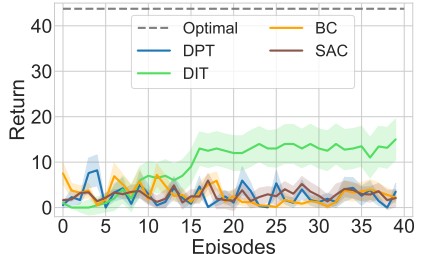 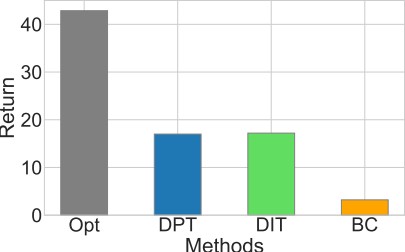 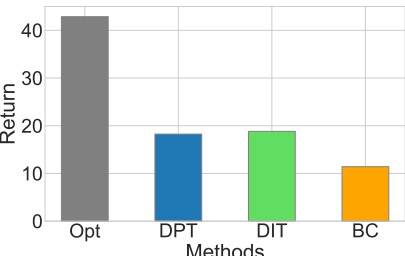

Figure 4: Ablation Study on **Miniworld**. From left to right: online testing, offline random, offline expert.

In the offline setting, DIT again demonstrates competitive performance with DPT. These results indicate that DIT has effectively leveraged the pretraining dataset to a significant extent.

## 6. Discussion

We have proposed a framework DIT for pretraining TMs from suboptimal historical data for ICRL. DIT has guaranteed policy improvements over the suboptimal behavior policies and demonstrated superior empirical performance on a comprehensive set of ICRL benchmarks. Despite these strengths, DIT still requires the behavior policies to have reasonable rewards. Most historical data typically adheres to this constraint. That said, it is highly unlikely to infer near-optimal actions solely from random trajectories without any information about optimal policies. To this end, we will further explore the limits of the proposed weighted pretraining framework in future work.

## Acknowledgements

Juncheng Dong and Vahid Tarokh were supported in part by the National Science Foundation (NSF) under the National AI Institute for Edge Computing Leveraging Next Generation Wireless Networks Grant #2112562. Ethan Fang is partially supported by NSF grants DMS-2346292 and DMS-2434666. Zhuoran Yang acknowledges support from NSF DMS 2413243.

## Impact Statement

This paper presents work whose goal is to advance the field of Machine Learning. There are many potential societal consequences of our work, none of which we feel must be specifically highlighted here.

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

## A. Results on Meta-World and Half-Cheetah

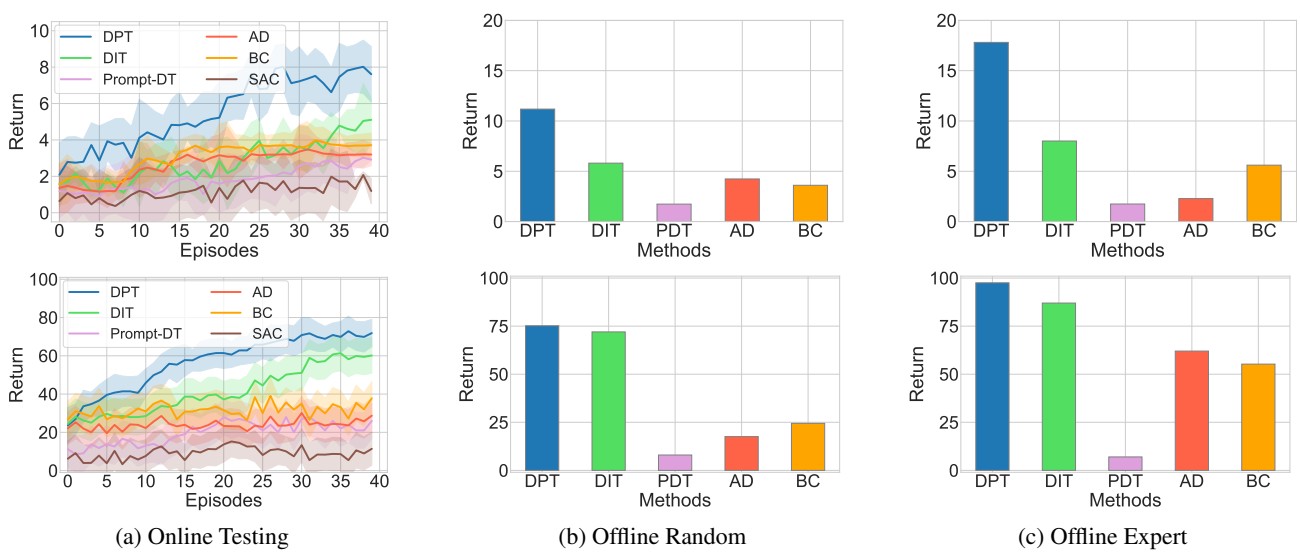

(a) Online Testing    (b) Offline Random    (c) Offline Expert

Figure 5: Top row shows results on **Meta-World**; bottom row shows results on **Half-Cheetah**.

## B. Baselines

### B.1. Bandit Algorithms

**Empirical Mean (EMP).**    We follow (Lee et al., 2024) to consider a strengthened version of EMP which, in the offline setting, only chooses from actions that have been observed at least once in the offline dataset while, in the online setting, at least choosing every action once. At every time step, EMP chooses actions as

$$\widehat{a} \in \arg\max_{a \in \mathcal{A}} \{\widehat{\mu}_a\},$$

where $\widehat{\mu}_a$ is the average observed reward for action $a$.

**Upper Confidence Bound (UCB).**    Motivated by the Hoeffding's Inequality, at each time step, UCB chooses actions as

$$\widehat{a} \in \arg\max_{a \in \mathcal{A}} \left\{\widehat{\mu}_a + C \cdot \sqrt{1/n_a}\right\},$$

where $C$ is a hyperparameter and $n_a$ is the number of times $a$ has been chosen. For unseen actions, $\widehat{\mu}_a$ is set to 0 and $n_a$ is set to 1. We follow (Lee et al., 2024) to set $C$ to be 1 as it demonstrates the best empirical performance.

**Lower Confidence Bound (LCB).**    LCB is on the contrary of UCB. In the offline setting, LCB only chooses from observed actions in the offline dataset. Specifically, it chooses actions as

$$\widehat{a} \in \arg\max_{a \in \mathcal{A}} \left\{\widehat{\mu}_a - C \cdot \sqrt{1/n_a}\right\},$$

where $C$ is a hyperparameter and $n_a$ is the number of times $a$ has been chosen. Similar to hyperparameter of UCB, the hyperparameter $C$ for LCB is also set to 1 due to its strong empirical performance.

**Thompson Sampling (TS).**    We use Gaussian TS (Russo et al., 2018) with a Gaussian prior. The mean and variance of the prior are set to the true mean and variance of the pretraining tasks: 0 for mean and 1 for variance.

## B.2. RL Baselines

**Decision-Pretrained Transformer (DPT).** The Decision-Pretrained Transformer (DPT) is designed to perform in-context learning for reinforcement learning (RL) tasks by leveraging a supervised pretraining approach. The core idea is to train a transformer model to predict optimal actions given a query state and a corresponding in-context dataset, which contains interactions from a variety of tasks. These interactions are represented as transition tuples consisting of states, actions, and rewards, offering context for decision-making. During pretraining, DPT samples a distribution of tasks. For each task $T_i$, an in-context dataset $D_i$ is constructed to include sequences of state-action-reward interactions that represent past experience with that task. Additionally, a query state $s^*$ is sampled from the MDP's state distribution, and the model is trained to predict the optimal action based on this query state and the context $D_i$. Formally, the training objective is to minimize the expected loss over the sampled task distribution by predicting a distribution over actions given the state and context.

**Prompt-based Decision Transformer (Prompt-DT).** Prompt-DT arranges its data to facilitate few-shot policy generalization by using trajectory prompts. For each task $T_i$, a prompt $\tau_i^*$ of length $K^*$ is constructed from few-shot demonstration data $P_i$, containing tuples of state, action, and reward-to-go $(s^*, a^*, \hat{G}^*)$. This prompt encodes task-specific context necessary for policy adaptation. Additionally, the recent trajectory history $\tau_i$ of length $K$, sampled from an offline dataset $D_i$, is appended to the prompt to form the full input sequence $\tau_{\text{input}}$. Formally, this input sequence is represented as $\tau_{\text{input}} = (\tau_i^*, \tau_i) = (\hat{r}_1^*, s_1^*, a_1^*, \ldots, \hat{r}_{K^*}^*, s_{K^*}^*, a_{K^*}^*, \hat{r}_{K^*+1}, s_{K^*+1}, a_{K^*+1}, \ldots, \hat{r}_{K^*+K}, s_{K^*+K}, a_{K^*+K})$. This sequence contains $3(K^* + K)$ tokens, following the state-action-reward format. The full sequence $\tau_{\text{input}}$ is then passed through a Transformer model, which autoregressively predicts actions at the heads corresponding to each state token. We follow Prompt-DT's setting and set $k = 20$.

**Algorithm Distillation (AD).** Algorithm Distillation (AD) transforms the process of reinforcement learning (RL) into an in-context learning task by training a transformer model to predict optimal actions based on a cross-episodic trajectory. AD gathers trajectories from training episodes, where each trajectory $T$ of length $H$ encodes the states, actions, and rewards observed over multiple episodes. Instead of training via traditional gradient updates, AD models the training history to predict actions for subsequent episodes, effectively distilling the behavior of RL algorithms like SAC into the transformer. This enables the model to learn directly from context, facilitating quick adaptation to new tasks and improving learning efficiency.

**Behavior Cloning (BC).** Behavior Cloning (BC) is a supervised learning approach for imitation learning, where the goal is to learn to mimic the behavior of a policy by mapping states to actions. Specifically, the objective is to minimize the discrepancy between the actions predicted by the learned policy $\pi_\theta$ and the target policy's actions, often through a loss function such as mean squared error or cross-entropy for continuous or discrete action spaces, respectively: $J(\theta) = \mathbb{E}_{(s_t, a_t) \sim D}[\ell(\pi_\theta(s_t), a_t)]$, where $D$ is the dataset of state-action pairs collected from the target policy's demonstrations, $s_t$ is the state at time step $t$, and $a_t$ is the corresponding target action.

**Soft Actor-Critic (SAC).** Soft Actor-Critic (SAC) is an off-policy deep reinforcement learning algorithm that balances exploration and exploitation by maximizing a trade-off between expected reward and entropy. The core objective of SAC is to learn a policy that not only maximizes cumulative rewards but also encourages exploration by maximizing the entropy of the policy's actions. SAC uses an actor network to predict actions, and two critic networks to estimate the Q-values of state-action pairs. The training objective involves learning the parameters of the policy to maximize a soft objective function: $J(\pi) = \sum_t \mathbb{E}_{(s_t, a_t) \sim D}[Q(s_t, a_t) - \alpha \log \pi(a_t|s_t)]$, where $Q(s_t, a_t)$ is the Q-value estimated by the critics, $\alpha$ is a temperature parameter controlling the trade-off between reward and entropy, and $\pi(a_t|s_t)$ is the action probability distribution given the state. SAC is trained by sampling mini-batches of transitions from a replay buffer to update the policy (actor) and Q-value estimates (critics). For model and training settings, we use the default implementation from Stable Baselines3 (Raffin et al., 2021).

## C. Theoretical Results

### C.1. Proof of Proposition 4.1

*Consider the following optimization problem where $\mathbb{E}_{\tau,s,a}$ is defined as in Equation (2) except that $a \sim \pi(a|s;\tau)$, i.e., the action is sampled from the task-conditioned policy rather than the behavioral policies:*

$$\max_{\pi} J(\pi) = \mathbb{E}_{\tau,s,a}\left[ \underbrace{A_\tau^b(s,a)}_{(I)} - \eta \cdot \underbrace{D_{\mathrm{KL}}(\pi(\cdot|s;\tau)\|\pi_\tau^b(\cdot|s))}_{(II)} \right], \tag{10}$$

*where $D_{\mathrm{KL}}$ is the Kullback–Leibler (KL) divergence, and let $\pi^\star \in \arg\max_\pi J(\pi)$ be its optimizer. Then we have for any policy $\pi(a|s;\tau)$,*

$$\mathbb{E}_{\tau\sim p(\tau),s\sim d_\tau(s)}\left[ D_{\mathrm{KL}}\left(\pi^\star(\cdot|s;\tau)\|\pi(\cdot|s;\tau)\right)\right]$$
$$= -\mathbb{E}_{\tau,s,a}\left[ \frac{1}{Z_\tau(s)} \exp\left(A_\tau^b(s,a)/\eta\right) \cdot \log \pi(a|s;\tau)\right] + C, \tag{11}$$

*where $\mathbb{E}_{\tau,s,a}$ is defined as in (2), $C$ is a constant independent of $\pi$ and $Z_\tau(s) = \sum_a \pi_\tau^b(a|s)\exp(A_\tau^b(s,a)/\eta)$.*

*Proof of Proposition 4.1.* For any task $\tau$ and fixed state $s$, we have

$$\max_\pi \mathbb{E}_{a\sim\pi(a|s;\tau)}\left[ A_\tau^b(s,a) - \eta \cdot D_{\mathrm{KL}}(\pi(\cdot|s;\tau)\|\pi_\tau^b(\cdot|s))\right]$$
$$= \min_\pi \mathbb{E}_{a\sim\pi(a|s;\tau)}[\log \frac{\pi(a|s;\tau)}{\pi_\tau^b(a|s)} - \frac{1}{\eta}A_\tau^b(s,a)]$$
$$= \min_\pi \mathbb{E}_{a\sim\pi(a|s;\tau)}\left[\log \frac{\pi(a|s;\tau)}{\pi_\tau^b(a|s)\exp(A_\tau^b(s,a)/\eta)}\right]$$
$$= \min_\pi \mathbb{E}_{a\sim\pi(a|s;\tau)}\left[\log \frac{\pi(a|s;\tau)}{\pi_\tau^b(a|s)\exp(A_\tau^b(s,a)/\eta)/Z_\tau(s)} - \log Z_\tau(s)\right]$$
$$= \min_\pi \mathbb{E}_{a\sim\pi(a|s;\tau)}\left[\log \frac{\pi(a|s;\tau)}{\pi_\tau^b(a|s)\exp(A_\tau^b(s,a)/\eta)/Z_\tau(s)}\right] \quad (Z_\tau(s) \text{ is independent of } \pi)$$
$$= \min_\pi D_{\mathrm{KL}}(\pi(\cdot|s;\tau)\|\pi_\tau^\star),$$

where $\pi_\tau^\star(a|s) = \pi_\tau^b(\cdot|s)\exp(A_\tau^b(s,a)/\eta)/Z_\tau(s)$. Note that the optimum $\pi$ for a fixed $s$ and task $\tau$ is obtained at $\pi = \pi_\tau^\star$, which is unique by the uniqueness property of KL divergence, i.e., $D_{\mathrm{KL}}(\pi\|\pi_\tau^\star) = 0$ if and only if $\pi = \pi_\tau^\star(a|s)$. Thus, the optimal task-conditioned policy is

$$\pi^\star(a|s;\tau) = \pi_\tau^\star = \pi_\tau^b(a|s)\exp(A_\tau^b(s,a)/\eta)/Z_\tau(s).$$

Thus, we further have

$$\mathbb{E}_{\tau\sim p(\tau),s\sim d_\tau(s)}\left[ D_{\mathrm{KL}}\left(\pi^\star(\cdot|s;\tau)\|\pi(\cdot|s;\tau)\right)\right]$$
$$= \mathbb{E}_{\tau\sim p(\tau),s\sim d_\tau(s),a\sim\pi^\star(a|s;\tau)}\left[\log \frac{\pi^\star(a|s;\tau)}{\pi(a|s;\tau)}\right]$$
$$= \mathbb{E}_{\tau\sim p(\tau),s\sim d_\tau(s)}\left[\sum_a \pi_\tau^b(a|s)\exp(A_\tau^b(s,a)/\eta)/Z_\tau(s)\log \frac{\pi^\star(a|s;\tau)}{\pi(a|s;\tau)}\right]$$
$$= -\mathbb{E}_{\tau\sim p(\tau),s\sim d_\tau(s),a\sim\pi_\tau^b(a|s)}\left[\exp(A_\tau^b(s,a)/\eta)/Z_\tau(s)\log \pi(a|s;\tau)\right] + C,$$

where $C = \mathbb{E}_{\tau\sim p(\tau),s\sim d_\tau(s),a\sim\pi_\tau^b(a|s)}\left[\exp(A_\tau^b(s,a)/\eta)/Z_\tau(s)\log \pi^\star(a|s;\tau)\right]$. $\qquad\square$

### C.2. Proof of Proposition 4.2

*Let $\pi^\star$ be the policy that optimizes Equation (3). For any task $\tau$ and policy $\pi$, let $G_\tau(\pi) = \mathbb{E}[\sum_{h=0}^\infty \gamma^h r_h|\pi,\tau]$ represent the expected reward of $\pi$ for $\tau$. Let $\pi_\tau^\star$ denote $\pi^\star(a|s;\tau)$. Then*

$$\mathbb{E}_{\tau\sim p_\tau}[G_\tau(\pi_\tau^\star) - G_\tau(\pi_\tau^b)] \geq \frac{\eta}{1-\gamma}\mathbb{E}_{\tau\sim p_\tau}[C_\tau^D] - \frac{2\gamma}{(1-\gamma)^2}\mathbb{E}_{\tau\sim p_\tau}\left[C_\tau^A\sqrt{C_\tau^D/2}\right], \tag{12}$$

where $C_\tau^D = \mathbb{E}_{s\sim d_\tau(s)}[D_{\mathrm{KL}}(\pi^\star(\cdot|s;\tau)\|\pi_\tau^b(\cdot|s))]$ and $C_\tau^A = \max_s |\mathbb{E}_{a\sim\pi^\star(a|s;\tau)}A_\tau^b(s,a)|$.

*Proof of Proposition 4.2.* First consider any fixed task $\tau$. From Corollary 1 in (Achiam et al., 2017), we have

$$G_\tau(\pi_\tau^\star) - G_\tau(\pi_\tau^b) \geq \frac{1}{1-\gamma}\sum_s d_\tau(s)\sum_a \pi^\star(a|s;\tau)A_\tau^b(s,a) - \frac{2\gamma C_\tau^A}{(1-\gamma)^2}\mathbb{E}_{s\sim d_\tau(s)}\|\pi^\star(\cdot|s;\tau) - \pi_\tau^b(\cdot|s)\|_{TV}, \quad (13)$$

where $C_\tau^A = \max_s |\mathbb{E}_{a\sim\pi^\star(a|s;\tau)}A_\tau^b(s,a)|$ and $\|\cdot\|_{TV}$ is the total variation distance between two distributions. In the proof of Propsition 4.1, we observe that: for any $\tau$ and $s$,

$$\pi^\star(\cdot|s;\tau) \in \arg\max_\pi \mathcal{L}(\pi,s) = \mathbb{E}_{a\sim\pi(a|s;\tau)}\left[A_\tau^b(s,a) - \eta\cdot D_{\mathrm{KL}}(\pi(\cdot|s;\tau)\|\pi_\tau^b(\cdot|s))\right].$$

Thus, $\mathcal{L}(\pi_\tau^\star, s) \geq \mathcal{L}(\pi_\tau^b, s)$, which implies that

$$\mathbb{E}_{a\sim\pi^\star(a|s;\tau)}\left[A_\tau^b(s,a) - \eta\cdot D_{\mathrm{KL}}(\pi^\star(\cdot|s;\tau)\|\pi_\tau^b(\cdot|s))\right] \geq \mathbb{E}_{a\sim\pi_\tau^b(a|s;\tau)}\left[A_\tau^b(s,a)\right] = 0.$$

Hence, we have

$$\mathbb{E}_{s\sim d_\tau(s),a\sim\pi^\star(a|s;\tau)}\left[A_\tau^b(s,a)\right] \geq \eta\mathbb{E}_{s\sim d_\tau(s)}[D_{\mathrm{KL}}(\pi^\star(\cdot|s;\tau)\|\pi_\tau^b(\cdot|s))]. \quad (14)$$

Moreover, from Pinsker's inequality (Canonne, 2022),

$$\mathbb{E}_{s\sim d_\tau(s)}\|\pi^\star(\cdot|s;\tau) - \pi_\tau^b(\cdot|s)\|_{TV} \leq \mathbb{E}_{s\sim d_\tau(s)}\sqrt{\frac{1}{2}D_{\mathrm{KL}}(\pi^\star(\cdot|s;\tau)\|\pi_\tau^b(\cdot|s))} \quad (15)$$

$$\leq \sqrt{\frac{1}{2}\mathbb{E}_{s\sim d_\tau(s)}[D_{\mathrm{KL}}(\pi^\star(\cdot|s;\tau)\|\pi_\tau^b(\cdot|s))]}, \quad (16)$$

where the last inequality comes from Jensen's Inequality. Pluging (15) and (14) into (13), we have

$$G_\tau(\pi_\tau^\star) - G_\tau(\pi_\tau^b) \geq \frac{\eta}{1-\gamma}\mathbb{E}_{s\sim d_\tau(s)}[D_{\mathrm{KL}}(\pi^\star(\cdot|s;\tau)\|\pi_\tau^b(\cdot|s))] \quad (17)$$

$$- \frac{2\gamma C_\tau}{(1-\gamma)^2}\sqrt{\frac{1}{2}\mathbb{E}_{s\sim d_\tau(s)}[D_{\mathrm{KL}}(\pi^\star(\cdot|s;\tau)\|\pi_\tau^b(\cdot|s))]}. \quad (18)$$

Taking expectation with respect to $\tau$ concludes the proof:

$$\mathbb{E}_{\tau\sim p_\tau}[G_\tau(\pi_\tau^\star) - G_\tau(\pi_\tau^b)] \geq \frac{\eta}{1-\gamma}\mathbb{E}_{\tau\sim p_\tau}[C_\tau^D] - \frac{2\gamma}{(1-\gamma)^2}\mathbb{E}_{\tau\sim p_\tau}\left[C_\tau^A\sqrt{C_\tau^D/2}\right], \quad (19)$$

where $C_\tau^D = \mathbb{E}_{s\sim d_\tau(s)}[D_{\mathrm{KL}}(\pi^\star(\cdot|s;\tau)\|\pi_\tau^b(\cdot|s))]$ and $C_\tau^A = \max_s |\mathbb{E}_{a\sim\pi^\star(a|s;\tau)}A_\tau^b(s,a)|$. $\quad\square$

**C.3. Justification for the identity $Z_\tau(s) = 1$**

Assume that $|A_\tau^b(s,a)/\eta| \ll |\log\pi_\tau^b(a|s)|$. Note that this can always be satisfied through reward normalization. Then

$$Z_\tau(s) = \sum_a \pi_\tau^b(a|s)\exp(A_\tau^b(s,a)/\eta) = \mathbb{E}_{a\sim\pi_\tau^b(a|s)}[\exp(A_\tau^b(s,a)/\eta)]$$

$$= \mathbb{E}_{a\sim\pi_\tau^b(a|s)}[1 + A_\tau^b(s,a)/\eta + o((A_\tau^b(s,a)/\eta)^2)] \quad \text{(by Taylor expansion)}.$$

Moreover, by definition of the advantage function, we have

$$\mathbb{E}_{a\sim\pi_\tau^b(a|s)}[A_\tau^b(s,a)] = \mathbb{E}_{a\sim\pi_\tau^b(a|s)}[Q_\tau^b(s,a)] - V_\tau^b(s) = 0.$$

Thus,

$$Z_\tau(s) = 1 + \mathbb{E}_{a\sim\pi_\tau^b(a|s)}[o((A_\tau^b(s,a)/\eta)^2)] \approx 1.$$

## D. MDP Environments

**Dark Room.** The agent is randomly placed in a room of $10 \times 10$ grids, and there is an *unknown* goal location on one of the grid. Thus, there are $10x10 = 100$ goals. The agent's observation is its current position/grid in the room, i.e., $\mathcal{S} = [10] \times [10]$. The agent needs to move to the goal location by choosing from 5 actions: to move in one of the 4 directions (up, down, left, right) or stay still. The agent receives a reward of 1 only when it is at the goal; otherwise, it receives 0. The horizon for Dark Room is 100. We follow (Lee et al., 2024) to use the tasks on 80 out of the 100 goals for pretraining, and reserve the rest 20 goals for testing our models' in-context RL capability for unseen tasks. The optimal actions are defined as: move up or down until the agent is on the same vertical position as the goal; otherwise move left or right until the agent reaches the goal.

**Miniworld.** The agent is placed in a room with four boxes of different colors, one of which being the target box. The goal is to reach a box of a specific color in the room. The agent receives a $(25 \times 25 \times 3)$ color image and its 2-D direction as input, and can choose from four possible actions: to turn left/right, move straight forward, or stay still. Similar to Dark Room, it receives a reward of 1 only when it is near the target box while the horizon is 50. The optimal actions are defined as follows: turn left/right towards the correct box if the agent's front is not within 15 degrees of the correct box; otherwise move forward and stay if the agent is near the box.

**Meta-World.** The agent needs to control a robotic arm to pick up an object and place it at a designated target location. In each task, the state space is in 39 dims including the gripper's position and state (open or closed), the 3D position of the object to be manipulated, and the coordinates of the target location. The agent operates in a continuous action space, where it can adjust the gripper's 3D position and control the open/close state to enable successful grasping and releasing of the object. It provides partial rewards for moving the gripper towards the object, grasping it correctly, transporting it to the target location, and successfully releasing it there. The task goal is to learn an optimal policy that efficiently achieves the sequence of actions required to pick up and accurately place the object at the specified location. Each task has a different goal position. We train in 15 tasks and test in 5 tasks.

**Half-Cheetah.** The agent needs to control a 2D half-cheetah robot to achieve and maintain varying target velocities, which change across episodes. The state space contains the cheetah's motion, including joint angles, velocities, body velocity, and position. These observations enable the agent to learn intricate movement patterns and maintain balance while running. The action controls the torques applied to each joint of the cheetah, thus dictating its locomotion and stability. The reward is designed to align with the core task objective: matching the agent's velocity to the target velocity. Each task has different target velocity, and we use 35 tasks to train and 5 to test.

## E. Pretraining Dataset

**Pretraining Datasets for Dark Room and Miniworld.** To ensure coverage of optimal actions (so that optimal policies can be inferred), at every step, with probability $p$ (respectively $1 - p$) we use optimal policy (respectively random policy) to choose action. We choose $p$ so that the average reward of the trajectories in the pretraining dataset is less than $30\%$ of that of the optimal trajectories, reflecting the challenging yet common scenarios. For Dark Room, to test whether DIT models can generalize to unseen RL problems in context, we collect context datasets from only 80 out of the total 100 goals and reserves the rest 20 for testing. For each training goal, we follow the setting of DPT to collect 1k context datasets, leading to a total of 80k context datasets in the pretraining dataset (64k for training and 16k for validation). For Miniworld, we collect 40k context datasets (32k for training and 8k for validation), 10k datasets for each of the four tasks corresponding to four possible box colors.

**Pretraining Datasets for Meta-World and Half-Cheetah.** We construct the pretraining datasets using historical trajectories generated by agents trained with *Soft Actor Critic* (SAC). Specifically, SAC is trained until convergence for each task, then we sample from its learning trajectories to build the dataset. Our SAC model training follows the settings outlined in (Haarnoja et al., 2018). For the Meta-World environment, we use its built-in deterministic policy as the optimal policy; for Half-Cheetah, we use the optimal SAC policy. In Meta-World, we used 15 tasks to train and 5 to test. Similarly, for Half-Cheetah, we used 35 tasks to train and 5 to test.

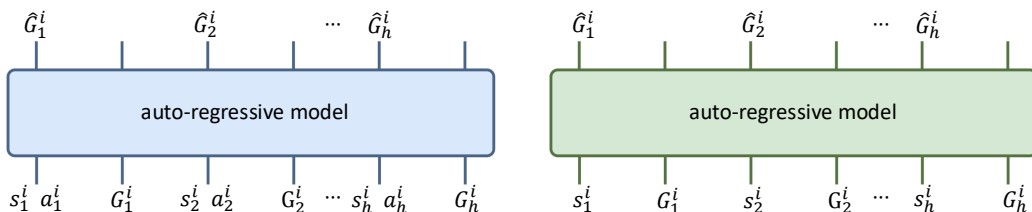

Figure 6: Model structure of the in-context action-value transformer $\widehat{Q}$ (left) and value transformer $\widehat{V}$ (right) on the trajectory of the $i$-th pretraining task.

## F. Training Parameters.

For all methods, we use the AdamW optimizer with a weight decay of $1e-4$, a learning rate of $1e-3$, and a batch size of 128.

## G. Model Details

**Decision Transformer Architecture.** Our model is based on a causal GPT-2 architecture (Radford et al., 2019). It consists of 6 attention layers, each with a single attention head, and an embedding size of 256. To separately encode state, action, and reward pairs, we employ three fully connected layers. We use a single fully connected layer to decode from the transformer's output.

**Value Function Transformer Architecture.** The architecture of the value function transformer mirrors that of the decision transformer.

## H. Computation Requirements

Our experiments can be conducted on a single A6000 GPU. It typically takes less than one hour to generate the required dataset for training in parallel. For PPO, training usually takes less than 10 minutes per task. For the other methods, we observe that the transformer model converges within 50 epochs.

## I. Pseudocodes

---
**Algorithm 1** Pretraining of Decision Importance Transformer

---
1: **Input:** Pretraining Dataset $\mathcal{D} = \{D^i\}$; transformer models $T_\theta, \widehat{Q}_\zeta, \widehat{V}_\phi$.
2: //   In-context Estimation of Advantage Functions
3: Randomly initialize and train $\widehat{Q}_\zeta$ and $\widehat{V}_\phi$ by optimizing the loss in Equation (8).
4: Construct the in-context advantage estimator as:

$$\widehat{A}_b = \widehat{Q}_\zeta - \widehat{V}_\phi.$$

5: //   Weighted Pretraining
6: Randomly initialize $T_\theta$.
7: With trained $\widehat{A}_b$ and $\mathcal{D}$, train $T_\theta$ by optimizing the loss in Equation (9).

---

## J. Estimation of Advantage function

Figure 7 illustrates the performance of our value function estimators. Notably, the ground truth labels represent the cumulative rewards empirically sampled using Monte Carlo, rather than the in-trajectory cumulative rewards. From the two graphs, we observe that our function estimator effectively learns the empirical distribution of cumulative rewards. Furthermore, the difference between the $Q$-function and $V$-function estimators provides the advantage function.

**Algorithm 2** Deployment of In-Context RL Models

1: **Input:** Pretrained transformer Model $T_\theta$; Horizon of episodes $H$; Number of episodes $N$ for online testing; Offline dataset $D_{\text{off}} = \{(s_h, a_h, s_{h+1}, r_h)\}_h$, consisting of transitions collected by a behavioral policy.
2: //   Offline Testing
3: **for** every time step $h \in \{1, \dots, H\}$ **do**
4:    Observe state $s_h$
5:    Sample action with $T_\theta$:

$$a_h \sim T_\theta\left(\cdot|s_h, D_{\text{off}}\right)$$

6:    Collect reward $r_h$
7: **end for**
8: //   Online Testing
9: Initialize an empty online data buffer $D_{\text{on}} = \{\}$
10: **for** every online trial $n \in \{1, \dots, N\}$ **do**
11:    **for** every time step $h \in \{1, \dots, H\}$ **do**
12:      Observe state $s_h$
13:      Sample action with $T_\theta$:

$$a_h \sim T_\theta\left(\cdot|s_h, D_{\text{on}}\right)$$

14:      Collect reward $r_h$
15:    **end for**
16:    Append the collected transitions $\{(s_h, a_h, s_{h+1}, r_h)\}_h$ into $D_{\text{on}}$
17: **end for**

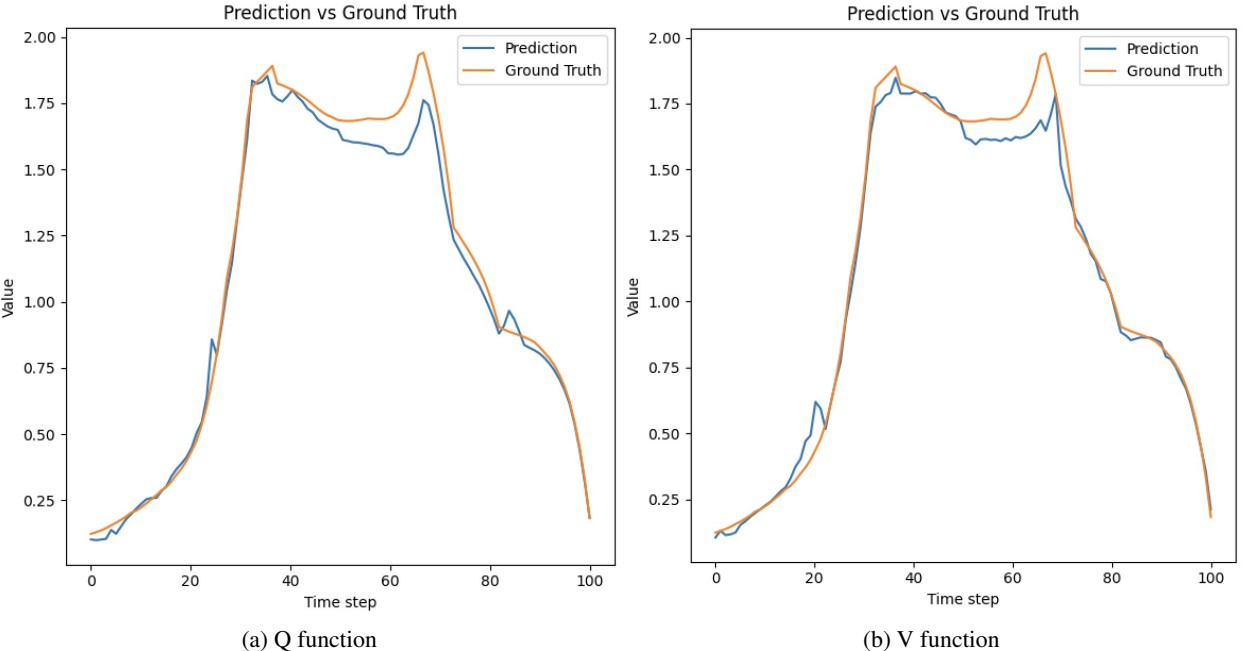

(a) Q function              (b) V function

Figure 7: Performance of Q and V function estimator. On the x-axis is time step of horizon; on the y-axis is the model predictions or ground truth values.

## K. Effectiveness of in-context trajectory

Figure 8 illustrates the effectiveness of the in-context trajectory for DIT. Since DIT predicts actions based on the current state and the historical states in the in-context trajectory, it is crucial to ensure that the task goal of the in-context trajectory aligns with the current task that DIT is predicting. Here, "In Task" refers to cases where the in-context trajectory is sampled

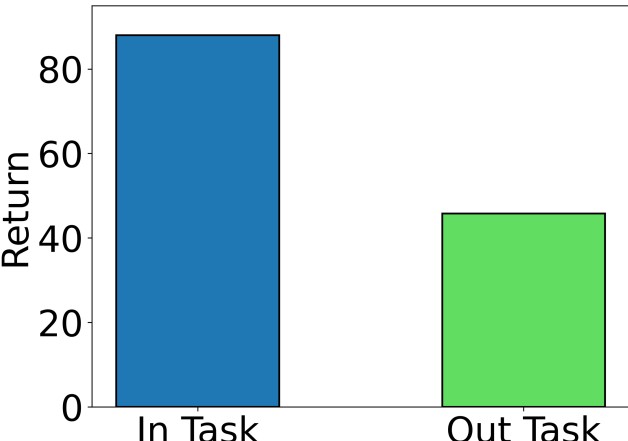

Figure 8: Performance of DIT when the in-context trajectory is aligned (In Task) or misaligned (Out Task) with the current task goal.

from the same task as the current task, while "Out Task" indicates that the in-context trajectory is sampled from a different task.

From Figure 8, we observe that alignment between the in-context trajectory and the current task goal is critical for effective performance. This finding also validates that DIT relies heavily on the in-context trajectory for action prediction, as misalignment with the current task goal leads to a significant decrease in performance.

