# OpenReview forum: "In-Context Reinforcement Learning From Suboptimal Historical Data"
_ICML.cc/2025/Conference — ICML 2025 poster_

### Official Review · Reviewer_g8M9 · 2025-03-10

**Overall Recommendation:** 2

**Summary:**

The paper introduces a method for multi-task RL meta-learning with suboptimal behavior policies. The goal is to train a common transformer model to imitate and improve upon the observed behaviors to maximize online rewards in new tasks. Towards this end, the paper introduces two methods: 1. a weighted policy imitation algorithm where the weights are based on exponentiated advantage function values and 2. a model architecture to transfer key dataset characteristics to infer the type of the new tasks. Experimental results on bandits and standard RL problems are included.

**Claims And Evidence:**

Yes. Claim 1 is supported by an equilibrium proof and Claim 2 is supported by a detailed model design. The authors provided empirical evidence to supplement the discussions.

**Essential References Not Discussed:**

Not to the best of my knowledge.

**Experimental Designs Or Analyses:**

Partially. The author provided detailed evaluation setups, but they missed to include a few baselines in bandit experiment. The authors should also consider policy reinforce algorithm in the RL experiments.

**Methods And Evaluation Criteria:**

Yes. The problem is typical of RL meta-learning and the datasets are standard for bandit / RL problems.

**Other Comments Or Suggestions:**

I like the paper a lot, but I cannot vouch for acceptance when there are still doubts in the experiments. To gain confidence, I would appreciate:
1. Completing Figure 2
2. Running Figure 3 & 4 until the proposed method meets optimal performance
3. Introducing policy reinforce baselines to show that simpler alternatives would not be sufficient to model reward values

**Other Strengths And Weaknesses:**

Strengths:
1. I like Proposition 4.1, which forms the foundation of the proposed policy improvements. The conclusions seem intuitive.
2. The follow-up algorithmic description is detailed, increasing my confidence in the empirical results.
3. The experiments validates the author's claims.

Weaknesses:
1. In the discussions of Proposition 4.1, the authors discarded the partition function. This could introduce additional variance, leading to empirical instability. The authors should consider other normalization techniques such as batch-normalization. The authors should also include closely-related policy reinforce baselines where the exponentiated advantage function is replaced with plain reward function.
2. A few results are missing from the bandit experiment, indicating a rush to complete the paper?
3. In the training details, for L_req objective, the authors expected the average of the q function to be the value function. This is only true for on-policy roll-outs. For off-policy roll-outs, the authors should also include policy-likelihood ratios in the L_reg objective. Related, the time-indices for the Bellman equations are in backwards.

**Questions For Authors:**

Regarding the three weaknesses:
1. Can the authors include and discuss policy reinforce methods in the experiments?
2. Can the authors include batch-normalization in the proposed methods - assuming that the authors left out normalization due to computational complexity?
3. Please complete Figure 2. Also for Figure 3 & 4, the methods are still far from converging to the optimal. How would you close this gap?

**Relation To Broader Scientific Literature:**

The authors were exhaustive in the related work section. The authors surveyed decision transformers (PDT), algorithmic distillation, and general behavior cloning. The authors appear less focused on meta-learning for RL, though the literatures there could be less a bit old.

**Theoretical Claims:**

Yes. The main theoretical claim is in Proposition 4.1. I briefly thought about it and the conclusions seem reasonable based on standard analysis on exponential family models.

---

> ### Author Rebuttal · Authors · 2025-04-01
>
> Thank you for your thoughtful and constructive feedback. We hope our responses below address your concerns.
>
> ### Experiments
>
> > **Bandit (Figure 2).** Our bandit experiments are designed to be comprehensive and aligned with established evaluation standards. As bandit problems are well-understood, **we follow the evaluation protocol used in DPT [1], comparing DIT against strong baselines such as the known optimal algorithms Thompson Sampling and UCB/LCB, as well as DPT itself** — which serves as an oracle for DIT due to its access to additional optimal bandit information during pretraining. These comparisons provide a rigorous evaluation of DIT in this setting.
>
> > **MDPs (Figures 3(a) & 4(a)).** The goal of these experiments is to evaluate adaptation speed — how quickly each method improves reward on new tasks. **While DIT has not fully converged to its optimal performance in Figure 3, the performances of all baselines have plateaued, and DIT already surpasses them in terms of performance**. We believe this clearly demonstrates the superior adaptability of DIT.
>
> > To provide additional empirical insights, we follow your advice to continue evaluation until the performance of DIT fully converges. As DIT’s performance in Figure 4(a) (Miniworld) has already converged, we focus on Darkroom and we run 10 additional episodes to see the final converged reward, it turns out that **DIT will continue advance the score a bit then keep stable (around 50), compared to 40 at episode 40**, expanding its performance gain over baseline methods.
>
> | Episode  | 5   | 10   | 15   | 20   | 25   | 30   | 35   | 40   | 41  | 42  | 43  | 44  | 45  | 46  | 47  | 48  | 49  | 50  |
> |----------|-----|------|------|------|------|------|------|------|-----|-----|-----|-----|-----|-----|-----|-----|-----|-----|
> | Reward   |3.87 |11.5 |16.9 |19.3 |23.8 |33.0 |36.9 |44.3 |45.5 |45.8 |46.2 |46.4 |46.8 |47.1 |47.7 |47.4 |47.2 |47.4 |
>
> ### Method
>
> > **Normalizing Constant and Policy Reinforce Baselines.** We prove in Appendix D.3 that, when using the exponentiated advantage function for reweighting, the normalizing constant (partition function) can be safely ignored and set to 1.
>
> > To further illustrate the effectiveness of DIT, we follow the reviewer’s suggestion to conduct an extra ablation study to compare DIT to several policy-reinforced baselines with different reweighting schemes. We evaluated three weight alternatives:
> > 1. **Cumulative reward**
> > 2. **Exponentiated cumulative reward**
> > 3. **Batch-normalized DIT**, where advantage values are first estimated and then normalized across all trajectories in the pretraining dataset.
>
> > We consider offline testing from expert trajectory on Meta-world and summarize the key results in the following table.
>
> | Method                                      | Cumulative Rewards | Exponentiated Cumulative Rewards | DIT (Batch-Norm Advantage) | DIT |
> |---------------------------------------------|--------------------|----------------------------------|-----------------------------|-----|
> | Return (higher means better)                                  | 5.0                | 4.7                              | 4.1                         | 8.2 |
>
>
> > We observe that **DIT significantly outperforms all baselines**. In particular, batch-normalized DIT performs worse than standard DIT. This **aligns with our theoretical analysis**, which highlights the importance of using trajectory-specific weights. Normalizing across the entire dataset violates this principle, which likely degrades performance.
>
> > We will include these results and a discussion of the policy reinforcement methods into our experiment section to further improve the quality of our work. We thank the reviewer for this constructive suggestion.
>
> ### Training Objective (on-policy value function)
> > Thank you for this insightful observation. We agree that the value function equals the expected Q-function only under on-policy rollouts.
>
> > Indeed, in our case, DIT performs pretraining using trajectories generated by behavioral policies, and the learned value functions are used to weight the actions taken by those same behavioral policies. **Thus, all training rollouts are on-policy with respect to the value functions being used, and there are no off-policy discrepancies in the objective**.
>
> > We believe this is one **key strength** of DIT and will clarify this point in the revised version. Additionally, we appreciate the note about the time indices in the Bellman equations and will correct them accordingly.
>
> [1] Lee, Jonathan, et al. "Supervised pretraining can learn in-context reinforcement learning." Advances in Neural Information Processing Systems 36 (2023): 43057-43083.

---

> > ### Comment · Reviewer_g8M9 · 2025-04-05
> >
> > I would not change my score, primarily because Figures 3a and 3b reveal a notable gap between the converged DIT policy and the Optimal policy. Although the proposed algorithm demonstrates relatively strong performance, it still suffers unrecoverable losses from learning with suboptimal demonstrations. I attribute this shortfall to the DIT algorithm’s separation of advantage function estimation and policy learning into two discrete steps. In contrast, a fully off-policy RL method that iterates these processes in tandem would likely provide a more complete solution.

---

> > > ### Author Response · Authors · 2025-04-05
> > >
> > > We appreciate your continued engagement and the thoughtful feedback regarding Figures 3a and 3b. We would like to respectfully clarify several points that we believe directly address this remaining concern.
> > >
> > > ---
> > >
> > > ### **The Challenge of Learning from Suboptimal Data and Generalizing In-Context to New Tasks**
> > >
> > > The primary goal of our work,  and ICRL more broadly, is to enable **fast adaptation to new RL environments** using only a **small number of demonstration trajectories**. This is **fundamentally different** from traditional offline RL, which assumes **abundant** suboptimal data collected from the **same environment** as the evaluation setting.
> > >
> > > We would like to highlight that **even in this simpler setting**, recovering an optimal policy from suboptimal data (e.g., in our experiments, only ~30% of optimal performance) is already extremely challenging, as recovering optimal policies requires sufficient state-action coverage within the offline data.
> > >
> > > Therefore, the performance gap observed in Figures 3a and 3b should  be viewed in the proper context of this work, characterized by:
> > > (a) **Learning from suboptimal data**, and
> > > (b) **Generalizing to unseen tasks with only a few adaptation trajectories.**
> > >
> > > Notably, even ICRL methods pretrained **with optimal actions** (DPT) exhibit a similar gap. Thus, we believe the residual gap is not a limitation of DIT’s architecture, but a natural outcome of the **inherent difficulty** of the problem setting.
> > >
> > > ---
> > >
> > > ### **Strong Empirical Results Despite the Challenges**
> > >
> > > As the reviewer notes, **DIT shows strong performance**, even in some cases outperforms those pretrained with optimal action labels, for example, in Figure 3a. We respectfully emphasize that this is a **highly non-trivial result**, especially considering that DIT is trained **exclusively** on suboptimal demonstrations.
> > >
> > > While perfect convergence to the optimal policy is desirable, **consistently outperforming prior methods** in this significantly harder setting is an important contribution. Furthermore, using only suboptimal data makes DIT more **practical and broadly applicable** to real-world scenarios where optimal demonstrations are unavailable.
> > >
> > > ---
> > >
> > > ### **Regarding the Tandem Alternative**
> > >
> > > We appreciate the reviewer’s suggestion regarding the benefits of fully off-policy algorithms that couple advantage estimation and policy learning. However, such methods often face well-known challenges, including instability, sensitivity to hyperparameters, particularly when trained on fixed and narrow suboptimal datasets.
> > > Moreover, applying such iterative updates in a transformer-based architecture adds substantial complexity and introduces its own set of optimization challenges.
> > >
> > > DIT, by contrast, provides a **simpler and more robust solution**, tailored for demonstration-based generalization without requiring access to optimal labels. We agree that combining DIT with more iterative learning mechanisms is a promising future direction, and we are enthusiastic about exploring this line of work. Still, we believe DIT’s current formulation already represents a **substantial and novel contribution**.
> > >
> > > ---
> > >
> > > ### **Final Remarks**
> > >
> > > In summary, DIT introduces a **new paradigm** for ICRL, specifically tackling the practical challenge of pretraining using only suboptimal offline data. It is **theoretically motivated** and achieves **strong empirical results** compared to relevant baselines within a **highly demanding setting** characterized by suboptimal data and few-shot generalization requirements.
> > >
> > > Considering the inherent difficulties of the problem, DIT's strong relative performance, its practical advantage (no need for optimal labels), and the thoughtful design choices addressing the specific challenges of ICRL, we kindly ask you to reconsider your evaluation, as we believe the paper offers significant contributions to the ICRL field.
> > >
> > > We are very willing to **add a discussion clarifying these points** (especially regarding the ICRL context vs. standard offline RL and the rationale behind DIT's design choices) to the manuscript to ensure the community fully benefits from our findings.
> > >
> > > Thank you once again for your valuable time and feedback.
> > >
> > > Sincerely,
> > >
> > > The Authors

---

### Official Review · Reviewer_GczK · 2025-03-12

**Overall Recommendation:** 4

**Summary:**

The new method Decision Importance Transformer (DIT) was proposed in the paper. This method is an enhancement of existing Decision Pretrained Transformer (DPT). While DPT requires expert target actions for training, DIT can be trained on trajectories sampled from suboptimal behavioral policies.  Since there is no need in expert policies, the proposed approach is easier to use and more versatile than DPT and other ICRL methods. To learn near-optimal policies through sub-optimal historical data, exponential reweighting with advantage function technique was utilised. Authors propose to add weight into common next-action prediction objective, to force model prioritise actions with higher advantage values.

DIT was tested on bandit and MDP problems. Dark Room and Miniworld was used as environments with discrete action spaces, Half-Cheetah and Meta-World was used as environments with continuous action spaces.  In all settings DIT showed performance competitive to DPT despite being pretrained without the optimal action labels.

## Update after rebuttal
I appreciate the authors’ rebuttal, which clarified my questions and addressed points that were previously unclear. The primary weakness of the paper was the insufficient explanation of the experiments, which left many aspects unclear to readers. This issue has now been resolved in the rebuttal. The idea of using the Advantage function to train DPT on suboptimal data is both novel and valuable for the ICRL area. In light of these improvements, I have raised my score.

**Claims And Evidence:**

The main claims of the paper is following:

1. DIT is able to learn in-context and adapt to unseen tasks while it is pertained on sub-optimal data *without the optimal action labels.*
2. DIT models demonstrate competitive performance to that of DPT

The authors evaluated DIT on bandits and several MDP problems (Dark Room, Miniworld, Half-Cheetah and Meta-World). Bandits, Dark Room and Miniworld is a classic set of ICRL benchmarks and it is shown in the paper that DIT emerges in-context learning on this environments, but I have concerns about presented results. The scores of DPT on Dark Room and Miniworld  is much lower than it was reported in the original paper. Therefore, the statement that DIT has performance comparable to DPT is inaccurate.

Moreover, in my opinion, it would be great to evaluate DIT on Key-to-Door and Watermaze environments as these problems are well-known and common benchmarks for ICRL. For example these benchmarks are used in AD (https://arxiv.org/pdf/2210.14215) and AD-epsilon (https://arxiv.org/pdf/2312.12275) papers. Without these established benchmarks, the set of evaluation tasks seem less comprehensive and insufficient.

It is good, that authors evaluated the method on continuous control tasks (Half-Cheetah and Meta-World), this makes the claims made in the submission stronger. But I spotted several inaccuracies. It is said that “Meta-World has 20 tasks in total, to evaluate our approach’s ability to generate to new RL tasks, we use 15 tasks to train and 5 to test.”, but Meta-World benchmark has 50 task in total. Why only 20 tasks were used and which task were used for training and evaluation, there are no information about it in the submission. Also I have concerns about presented scores on Meta-World tasks. The values of returns are in [0, 10] segment, but experts on these tasks can achieve much more higher results ( hundreds if the length of episode is 100 as it was in JAT or thousands if the length of episode is 250 as it was in GaTo (https://arxiv.org/abs/2205.06175)), so the value of returns in the segment of [0, 10] seem like totally random agents and does not support the main claims.

**Essential References Not Discussed:**

Talking about “*stringent requirements on the pretraining datasets”* authors claim that AD need full learning trajectories of RL agents, but they don’t mention, that it is possible to collect dataset via noise distillation. This method is called $AD^{\epsilon}$ (https://arxiv.org/pdf/2312.12275). This work should be mentioned.

Also advantage weighted regression (https://arxiv.org/abs/1910.00177) is mentioned in this work, but there are no reference.

Authors are talking about using AWR in In-Context RL setting, but don’t mention AMAGO-2 (https://arxiv.org/pdf/2411.11188), where AWR was used. In my opinion, this work should be mentioned too.

**Experimental Designs Or Analyses:**

In my opinion experimental design and analysis is valid for this work.  All my concerns about experimental setup were considered in “Claims And Evidence” and “Methods And Evaluation Criteria” paragraphs.

**Methods And Evaluation Criteria:**

The proposed benchmarks is fully suitable for verifying the article's claims, but it could be enhanced by additional ones commonly-used in ICRL area.

Presented results is questionable and the setup of experiments are not fully described. What Meta-World tasks were chosen for training and test sets, has hyperparameters search been performed for considered methods, why demonstrated scores for DPT don’t match with reported values in original work, what is the length of context in pretraining datasets for each benchmark. In my opinion this information should be provided for better understanding the experiment setup and results of the work.

**Other Comments Or Suggestions:**

I do not have any other comments or suggestions.

**Other Strengths And Weaknesses:**

Strengths:

- The paper is well written and it's easy to follow.
- Detailed and clear description of the proposed method;
- Applying method from Offline RL into In-Context RL area is good and novel idea;
- Theoretical justification of the proposed method

Weaknesses:

- The experimental description is incomplete, leaving questions after reading the paragraph;
- The claimed DPT scores differ from those in the original article

**Questions For Authors:**

1. Why do the DPT returns on Dark Room and Miniworld differ between the submission and the original paper?
2. What Meta-World tasks were chosen for training and test sets?
3. In the mathematical expectation operator at the start (Expression 10), actions are sampled from the behavior policy for a certain task,  $a \backsim \pi_{\tau}^b(a|s)$. However, in the “Proof” section, the mathematical expectation operator samples actions from the meta-policy,  $a \backsim \pi(a|s; \tau)$. Why?
4. Did you check what DIT do on Meta-World tasks (render video) or what the expert return is on these tasks? According to my experience the good returns on Meta-World tasks (when agent actually solve the problem or almost solve the problem) are much higher than 10.
5. What is the length of the context in the pertained datasets?

I may change my evaluation and raise the score if the answers remove my concerns.

**Relation To Broader Scientific Literature:**

Decision Importance Transformer (DIT) is a novel approach in the field of In-Context Reinforcement Learning. Existing approaches, such as DPT(https://arxiv.org/pdf/2306.14892) and AD(https://arxiv.org/pdf/2210.14215), require expert data in their training datasets. However, collecting such data can be difficult, whereas suboptimal trajectories are only available for training agents. Many methods in offline RL, including CRR(https://arxiv.org/abs/2006.15134), IQL(https://arxiv.org/pdf/2110.06169), and CQL(https://arxiv.org/abs/2006.04779), utilize suboptimal data to learn better policies, yet no comparable techniques exist for ICRL. Advantage weighting is a well-known strategy in offline RL, and the authors propose a way to generalize it for ICRL.  Some works have begun exploring this—for example, AMAGO-2 (https://arxiv.org/abs/2411.11188) — but it remains an understudied area. In my opinion, this is an impactful contribution that will reduce the data requirements needed to train ICRL agents.

**Theoretical Claims:**

There are a few theoretical claims in the submission. I have carefully reviewed them and would like to ask some questions as I found contradiction. In Proposition 4.1, the optimization problem  $J(\pi)$ is considered. This objective is the expected value of the difference between the advantage function and the KL divergence.

In the mathematical expectation operator at the start (Expression 10), actions are sampled from the behavior policy for a certain task,  $a \backsim \pi_{\tau}^b(a|s)$. However, in the “Proof” section, the mathematical expectation operator samples actions from the meta-policy,  $a \backsim \pi(a|s; \tau)$.

From my understanding, the second variant is correct because if the expectation is taken over the behavior policy  $\pi_{\tau}^b(a|s)$, then maximizing  $J(\pi)$ becomes trivial. This happens since only the  $D_{KL}(\pi(.|s;\tau)||\pi_{\tau}^b(.|s))$ term would depend on $\pi(a|s; \tau)$.

---

> ### Author Rebuttal · Authors · 2025-04-01
>
> Thank you for your constructive comments. Please see our responses below to address your concerns.
>
> ### References
> > We appreciate these shared references. We will include a discussion of AD-\epsilon, AWR, and AMAGO-2 into our manuscript. In particular, we highlight that AD-\epsilon still requires sampling actions from the good policies to perform noise distillation. Moreover, to create noise distilled dataset, they need to actively sample trajectories following a noise schedule. **In comparison, our work assumes a setting of only historical data, which is more realistic and easier to satisfy**.
>
> ### Theoretical Results
> > The authors deeply appreciate the reviewer’s detection of this typo, and we will correct it in our updated manuscript. Yes, the expectation in Equation (3) should be with respect to the meta-policy $\pi(a|s\tau)$ and the expectation in Equation (4) is with respect to the behavioral policy $\pi^b_{\tau}(a|s)$: we use data collected by the behavioral $\pi^b_{\tau}(a|s)$ to learn a meta-policy  $\pi(a|s;\tau)$.
>
> ### Experiment Setup
> >**Comprehensive Benchmarks.** Our goal is to provide a more comprehensive empirical evaluation than prior works, such as DPT (which focuses on bandits and navigation tasks like Dark Room and Mini-world) and prompt-DT (which focuses solely on continuous control). **To that end, we include representative benchmarks from all three domains: bandits, navigation, and continuous control**.
> While we agree that Key-to-Door is a valuable benchmark, it shares significant structural similarities with tasks like Dark Room and MiniWorld. As such, we believe it would offer limited additional insight beyond the settings already covered. Due to the time constraints of the rebuttal phase, we are unable to include new results at this stage. However, **we are more than happy to run additional experiments on Key-to-Door and include them in the final manuscript to further strengthen the empirical evaluation**.
>
> > **Meta-World Benchmark.** We follow the same environment setting of Prompt-DT[1] to use 20 tasks from the Meta-World reach-v2 suit (specifically ML1-pick-place-v2), while we choose the task horizon to be 100. We choose the converged SAC policy as the optimal policy. The cumulative reward of a random policy is around 1.5 within a horizon of 100 and around 20 for the optimal policy. We will update the manuscript to make this clear.
>
> ### Extra Experiments
> > To provide additional insights, we conduct extra experiments to compare DIT with several policy reinforce [2] baselines where the reweighting is based on cumulative reward rather than the advantage function. Specifically, we consider reweighting with cumulative rewards and exponentiated cumulative rewards for offline testing from expert trajectory on Meta-world and summarize the key results in the following table.
>
> | Method                        | Cumulative Rewards | Exponentiated Cumulative Rewards | DIT |
> |------------------------------|--------------------|----------------------------------|-----|
> | Return (higher means better) | 5.0                | 4.7                              | 8.2 |
>
> > **The significantly improved performance of DIT over these two baselines further proves the effectiveness of our proposed method**.
>
> ### Performance of DPT.
>
> > The difference in DPT's performance between our work and the original paper stems from the nature of the pretraining datasets. In the original DPT paper, pretraining trajectories were collected using **uniformly random policies**, which ensured broad coverage of the state-action space. In contrast, as noted in our manuscript, we use suboptimal policies (achieving only ~30% of optimal performance) to collect pretraining data. This choice naturally reduces coverage and impacts DPT’s performance.
>
> > However, we believe **it better reflects realistic scenarios**—especially in practical applications where historical data are typically collected by suboptimal or heuristic policies, not uniformly random ones.
>
> > To ensure a fair comparison, both DPT and our method (DIT) are trained on the same set of suboptimal trajectories. However, **we provide DPT with additional privileged information**: a set of randomly sampled out-of-trajectory query states with corresponding optimal action labels sampled from optimal policies. This positions DPT as a strong oracle-style upper bound relative to DIT, further underscoring the strength of DIT’s performance under more constrained assumptions.
>
> [1] Xu, M., Shen, Y., Zhang, S., Lu, Y., Zhao, D., Tenenbaum, J., & Gan, C. (2022, June). Prompting decision transformer for few-shot policy generalization. In international conference on machine learning (pp. 24631-24645). PMLR.
>
> [2] Williams, R. J. (1992). Simple statistical gradient-following algorithms for connectionist reinforcement learning. Machine learning, 8, 229-256.

---

> > ### Comment · Reviewer_GczK · 2025-04-03
> >
> > Thank you for the clarification! It’s much clearer to me now. In my opinion, these details should be included in the paper to help readers better understand the work.
> >
> > Have you tried training DIT on trajectories collected using uniformly random policies, similar to what was done in the original DPT paper?

---

> > > ### Author Response · Authors · 2025-04-03
> > >
> > > We are gratified to learn that our previous response was helpful, and we will for sure incorporate the suggested details into the final manuscript.
> > >
> > > We indeed have tried DIT with uniformly random behavior policies. However, we encountered significant challenges. The reason is that the performance of these random policies was less than 1% of the optimal policies, rendering them unsuitable for providing reliable pretraining signals. This ineffectiveness is expected, and, as highlighted in our conclusion, we believe inferring near-optimal actions from purely random trajectories, devoid of information regarding optimal policies, is improbable. Furthermore, real-world historical data, as collected by companies, is more likely to originate from suboptimal policies (e.g., achieving 30% of optimal performance) than from uniformly random policies, which are rarely observed in practical scenarios.
> > >
> > > We hope this response can address your remaining concern.

---

### Official Review · Reviewer_EYEv · 2025-03-13

**Overall Recommendation:** 3

**Summary:**

This work focuses on in-context reinforcement learning where the source data/policy is suboptimal. In this case, the traditional ICRL algorithm could perform bad. This work proposes Decision Importance Transformer (DIT), which emulates the actor-critic algorithm in an in-context manner. It achieves superior performance than the baselines when the offline data is suboptimal.


## update after rebuttal

I appreciate the authors' rebuttal. However, I think the authors should address my Question 7 more properly in the paper. I am not fully convinced by the authors' explanation. DIT uses the state from the context as the query state while DPT considers a more general way (use a random query state), shouldn't DPT work better in terms of learning a general understanding of an MDP? DIT will be limited by the size and richness of the context dataset. In any case, it remains unclear why DIT can outperform DPT.

**Claims And Evidence:**

Overall yes. But I still have some questions and recognize some weaknesses. Please see them in the corresponding section below.

**Essential References Not Discussed:**

The following reference is missing, which also studies how the offline data affect the performance of ICRL.

Zisman, I., Kurenkov, V., Nikulin, A., Sinii, V., & Kolesnikov, S. (2023). Emergence of in-context reinforcement learning from noise distillation. arXiv preprint arXiv:2312.12275.

**Experimental Designs Or Analyses:**

Yes I checked. Please see the weakness and my questions below.

**Methods And Evaluation Criteria:**

Yes. The problem is useful and the benchmark environments are reasonable.

**Other Comments Or Suggestions:**

1. The authors are encouraged to not write equations in the text, which negatively impacts readability, e.g., the texts after Eq.(2).

2. Immediately introduce $\eta$ when first mentioning it in Eq.(2).

**Other Strengths And Weaknesses:**

1. In the sentence: "When presented with a context dataset containing environment interactions collected by unknown and often suboptimal policies, pretrained TMs predict the optimal actions for current states from the environmental information within the context datase". I have two concerns.

a) It sounds like there needs to be a pre-collected context dataset that is presented to the TMs. However, the online deployment of DPT already shows that the pretrained TMs can collect its own contexts online.

b) "predict **the optimal actions**" doesn't sound good to me, because AD does not predict the optimal actions, it predicts whatever in the source RL algorithm. Indeed, here the pretrained TMs predicts whatever it learns during the pretraining, which may not be the optimal actions, depending on which ICRL algorithm is considered. Maybe it is just a language thing, but make it more clear and accurate.

2. " For instance, large companies often maintain extensive databases of historical trajectories from non-expert users." Why? is there a reference? The authors can explain the motivation in a more prefessional way, in terms of the language and example.

3. In Line 138, "causal transformer" may not be familiar for the readers. Explain it with more details and provide the reference.

4. In section 3, the authors use $p_\tau$ to denote both pretraining and test distributions. If DIT is deployed in an unseen new test task, in general, the test distribution should differ from the pretraining distribution?

5. The sentence "$D_{off}$ contains trajectories gathered from a random policy in $\tau$" is not correct right? The offline deployment in DPT paper considers two offline datasets. The offline dataset can be either collected by random policy or the expert policy. See Figures 4 and 5 in DPT paper.

6. The weight in DIT is manually selected. Is there a way to automatically select/learn the weight? Or is there a hint that what weight should one choose given a problem?

7. In DPT's online deployment, how does it work initially when $D_{on}$ is empty? I mean, it appends a trajectory every episode. In the whole first episode (not just the first step), the context is empty. Does TM still predict actions?

8. The authors claim that "DIT learns to infer near-optimal actions from suboptimal trajectories" and "DIT is comparable to DPT in both online and offline testings, despite being pretrained without optimal action labels." These are very strong claims. My question is: How "suboptimal" is your dataset? Should the suboptimal trajectories contain the near-optimal actions? If so, how much it needs? Please explain it more explicitly and with more contents in the Introduction, **not just one or two sentences in the experimental section**. Otherwise readers might think that DIT learning from a very very terrible offline dataset can achieve comparable performance with DPT, which is irrealistic.

9. I am confused about the pretraining dataset for Meta-World and Half-Cheetah. If they are collected from SAC, how is that "suboptimal"?

10. I appreciate that the authors provide the codes for validation. However, there is no instructions at all about how to install the dependencies and how to run and evaluate the algorithms.

11. Why AD is not considered in Miniworld?

**Questions For Authors:**

1. Why in-context reinforcement learning works? I mean, after the Transformers are pretrained, they are kept frozen when applying to the **new unseen tasks**. Why it can learns in-context? Specifically, why the performance can improve when there is more and more contexts while the Transformers parameters are frozen?


2. This work is built upon DPT. But why DPT can learn in-context? As discussed in the original AD paper, the methods that learn an good policy like DT cannot improve in-context with frozen Transformers parameters, i.e., cannot reinforcement learn in context. In this sense, isn't DPT also learning optimal policies? Why DPT can show in-context reinforcement learning abilities?


3. Comparing with AD/DPT, the DIT proposed in this work will need two extra transformer-based value functions. Does it cost a lot? I think the authors should report how much cost does it need compared with the baseline in terms of e.g., running time? More interestingly, does DIT still perform well compared to the baselines when **they have the same time budget**? This comparison is important.

4. Why the authors want to select a expotential weight? What is the advantage?

5. The context is genenrally just {s, a, r, s'}. But in Q and V estimators, the context is {s, a, G}. Why do we want to use this? Is it fair that the label for Q and V transformers should be the return G. But why the context needs to contain G?

6. Which tasks do the authors consider in Meta-World?

7. It is **extremely surprising** that DIT performs better than DPT in Figure 3(a) and Figure 4(a). Figure 5 looks more normal where the DPT is the oracle baseline. In addition, why DPT in Figures 3 and 4 does not have the comparable performance as that in the original DPT paper?

**Relation To Broader Scientific Literature:**

N/A

**Theoretical Claims:**

Yes, they look correct to the best of my knowledge.

---

> ### Author Rebuttal · Authors · 2025-04-01
>
> Thank you for your helpful and constructive comments. We will incorporate the referenced work into the final manuscript. In addition, we will include citations supporting the availability of abundant historical data in real-world settings (e.g., using [1] as a standard reference). Please see below for our responses to your other concerns.
>
> ### Why DPT and ICRL Work
> > As discussed in the original DPT paper, on a high level, DPT internally conducts posterior sampling (PS) for MDP, where the transformer model is pretrained to infer the target MDP from the given context dataset and to take action according to the optimal policy for the inferred target MDP. **As the context size increases, the inferred MDP becomes closer to the true target MDP, thus leading to improved performance**. Meanwhile, there are also theoretical works establishing the efficacy of the supervised pretraining approach taken by DPT, e.g., [2].
>
> > **Prediction without a context dataset.** The DPT model can still predict an action using only the query state as input (without a context dataset), as the prediction is only based on the token corresponding to the query state.
>
> ### Method
>
> > **Choice of hyperparameter.** As shown in our theoretical result, $\eta$ represents the penalty for the KL divergence. Thus, if the historical data is less suboptimal (high quality), it should be high as we would like to stay close to the behavior policies; on the contrary, if the data is low quality, it should be chosen as a small value so that we can improve more over the behavior policies.
>
> > **Value transformer model structure.**  Our design follows the in-context learning setup, where transformers are trained for regression tasks $y = f(x)$ using sequences like $x_1, y_1, ..., x_n, y_n$. In our case, value estimations (for V and Q) can be seen as regression problems, e.g., set $x = s$ and $y = G$ (cumulative reward) when estimating V. Based on this, we design our value transformer as shown in Figure 6. Figure 7 provides empirical evidence that it can accurately estimate value functions in context.
>
> > **Cost of training in-context advantage estimator.** As detailed in the manuscript, we train them with standard supervised training objectives for transformers, which are stable and straightforward to optimize. While this moderately increases the total pretraining time, we emphasize that **the primary bottleneck in ICRL lies in collecting high-quality pretraining data, not in training time**. To this end, DIT reduces this data requirement considerably, trading off for a modest increase in training complexity. We believe **this tradeoff is worthwhile and offers strong practical benefits**, making ICRL more accessible and scalable in real-world applications.
>
> ### Experiments
> > **Historical Data Suboptimality.** In our experiments, we use behavioral policies with performance 30%-50% of the optimal policies to collect historical data, a commonly used setting for offline RL. We would like to clarify that we use the intermediate training checkpoints of SAC to collect trajectories so that they are indeed suboptimal. We will update the manuscript to make this clear in the introduction to avoid confusion.
>
> > **Experiment Setup.** For Meta-world, we follow the setting of Prompt-DT to use Meta-World reach v2. We use Mini-world mainly for ablation study to understand the importance of the proposed reweighting mechanism and the effect of lack of optimal action labels. Given this purpose, we compare DIT with DPT and a variation of DIT without reweighting.
>
> > **Performance of DPT.** The observed performance difference of DPT between this work and the original paper is due to the difference of the pretraining datasets. Due to space constraint, please refer to our response to Reviewer Gczk for more details.
>
> ### DIT sometimes outperforms DPT
> > We appreciate the reviewer’s insightful observation. This is indeed a compelling research direction.
>
> > At a high level, DPT uses a single randomly sampled query state paired with an optimal action label for each trajectory during pretraining, whereas DIT leverages multiple reweighted suboptimal action labels. **Interestingly, these many reweighted suboptimal labels can collectively provide a stronger learning signal than a single optimal label, resulting in better pretraining objectives in some settings**. Of course, if DPT were given access to many optimal action labels per trajectory, it would likely surpass DIT, as DIT does not use any optimal supervision.
>
> [1] Levine, S., Kumar, A., Tucker, G., & Fu, J. (2020). Offline reinforcement learning: Tutorial, review. and Perspectives on Open Problems, 5.
>
> [2] Lin, Licong, Yu Bai, and Song Mei. "Transformers as decision makers: Provable in-context reinforcement learning via supervised pretraining." arXiv preprint arXiv:2310.08566 (2023).

---

> > ### Comment · Reviewer_EYEv · 2025-04-03
> >
> > Thank you for your response. But please reply to my comments/questions one by one (like below) so that I can match your answers to the comments.
> >
> > Reviewer comments "*xxxxx*"
> > - Author response: "xxxxx"
> >
> > I proposed 20 comments/questions. It looks like that not all of my comments are addressed.

---

> > > ### Author Response · Authors · 2025-04-03
> > >
> > > We appreciate the reviewer’s follow-up question, and we are more than happy to address them following the reviewer’s requested template, which we couldn’t complete mainly due to the space constraint of a single response.
> > >
> > > > ### Questions in *Other Strengths and Weaknesses*
> > >
> > > **Answer to Q1.** We will follow the reviewer’s suggestion to fix this sentence, avoiding potential misunderstanding.
> > >
> > > **Answer to Q2 on references and motivations for companies possessing historical data.** In our ongoing era of Big Data, large companies, particularly in areas like e-commerce, navigation, ride-sharing, health care and online gaming, often collect and maintain users’ historical data. These stored historical data can be employed for various purposes: to improve user experience, to optimize service delivery, to detect anomalous behaviors, etc. While offline RL surveys and papers are often good references, please see below two example references
> > > - Chen, X., Wang, S., McAuley, J., Jannach, D., & Yao, L. (2024). On the opportunities and challenges of offline reinforcement learning for recommender systems. ACM Transactions on Information Systems, 42(6), 1-26.
> > > - Shi, T., Chen, D., Chen, K., & Li, Z. (2021). Offline reinforcement learning for autonomous driving with safety and exploration enhancement. arXiv preprint arXiv:2110.07067.
> > >
> > > **Answer to Q3.** We will include a detailed introduction. Thank you for this helpful suggestion.
> > >
> > > **Answer to Q4 and Q5 on testing task distribution and initial context dataset.**  While the testing task distribution for ICRL during deployment can be different to the pretraining task distribution, we assume them to be the same for simplicity, as it is not related to the key contribution of this work and all common benchmarks for ICRL follow this assumption. The context dataset indeed can contain trajectories collected from policies other than the random ones. We will update the manuscript to make this point clear.
> > >
> > > **Answer to Q6 on how to select weights for DIT.** Please see the **Choice of Hyperparameter** paragraph (in the first response).
> > >
> > > **Answer to Q7.** Please see the **Prediction without a context dataset** paragraph.
> > >
> > > **Answer to Q8 and Q9.** Please see the **Historical Data Suboptimality** paragraph.
> > >
> > > **Answer to Q10 on implementation.** Thank you for mentioning this point. Indeed, we plan to open-source all the implementations with full instructions for installation, training, and evaluation, after the publication of this paper.
> > >
> > > **Answer to Q11 on mini-world experiments not including AD.** We use Mini-world mainly for ablation study to understand the importance of the proposed reweighting mechanism and the effect of lack of optimal action labels. Given this purpose, we compare DIT with DPT and a variation of DIT without reweighting.
> > >
> > > ### Questions in *Other Comments Or Suggestions*
> > >
> > > We appreciate the reviewer’s suggestions and we will improve these two points in the final manuscript.
> > >
> > > ### Questions in *Questions for Authors*
> > >
> > > **Answers to Q1 and Q2.** Please refer to the **Why DPT and ICRL Work** section in the first response. Additionally,  regarding why DT cannot learn in-context, as stated in the Related Work section of the AD paper, “*Importantly, these prior methods use contexts substantially smaller than an episode length, which is likely the reason in-context RL was not observed in these works.*”
> > >
> > > **Answers to Q3.** Please see the **Cost of Training in-context advantage estimator** paragraph.
> > >
> > > **Answers to Q4 on advantage function and exponential weighting.** The advantage function tells how much better (or worse) a specific action is compared to the average action you would normally take in a given state. Thus, we use it to evaluate whether an action is good or bad. We choose exponential weight because, as shown in our theoretical results, it leads to guaranteed performance improvement.
> > >
> > > **Answers to Q5 on the design of value transformers.** Please see the **Value Transformer Model Structure** paragraph.
> > >
> > > **Answers to Q6 on the tasks used in Meta-world.** For Meta-world, we follow the setting of Prompt-DT to use Meta-World reach v2.
> > >
> > > **Answers to Q7.** Please see the **DIT sometimes outperforms DPT** and **Performance of DPT**  paragraphs in the first response.

---

### Official Review · Reviewer_JvGg · 2025-03-14

**Overall Recommendation:** 3

**Summary:**

The paper proposes the Decision Importance Transformer (DIT), a novel framework for in-context reinforcement learning (ICRL) that is designed to work with historical datasets generated by suboptimal behavioral policies. Unlike previous approaches that require optimal action labels or complete learning histories, DIT uses only suboptimal data. Its key idea is to incorporate an exponential reweighting scheme in the supervised pretraining objective. This weighting is derived from an estimated advantage function that is computed via transformer-based value and action-value estimators. The overall system is built upon an autoregressive transformer (using a GPT-2 backbone) that, once pretrained on a diverse set of tasks, can generalize to unseen tasks by extracting task-specific information from context trajectories. The paper supports its contributions both theoretically—with propositions that connect the weighted maximum likelihood objective to policy improvement—and empirically, by demonstrating competitive or superior performance on a range of problems (from bandit settings to challenging MDPs including navigation and continuous control tasks).

**Claims And Evidence:**

Core Claims:
- Policy Improvement via Reweighting: The authors claim that by reweighting action labels according to an estimated advantage function, the transformer can learn to “steer” suboptimal behavioral data toward near-optimal policies.
- Generalization from Suboptimal Data: DIT is posited to work well even when the training data do not contain optimal action labels, a scenario that is common in real-world settings where only historical data are available.

Evidence:
- Theoretical Analysis: The paper presents Proposition 4.1 and Proposition 4.2, which relate the weighted supervised objective to a policy improvement problem and provide conditions under which the learned policy strictly improves over the behavior policy. Although proofs are deferred to the appendix, the propositions outline a clear link between the exponential weighting scheme and performance guarantees.
- Empirical Results: Extensive experiments are conducted on both bandit problems and various MDPs (including navigation tasks like Dark Room and Miniworld, as well as continuous control tasks such as Meta-World and Half-Cheetah). The results show that DIT can match or exceed the performance of baselines (including methods that have access to optimal action labels) in both online and offline settings.

**Essential References Not Discussed:**

N/A

**Experimental Designs Or Analyses:**

The paper evaluates DIT on both simple (linear bandit) and complex (MDP) environments.
For bandit problems, the method is compared against theoretically optimal algorithms (e.g., UCB and Thompson Sampling), showing that DIT quickly identifies the optimal bandit.
In MDP settings, the authors test on environments with both sparse rewards (Dark Room, Miniworld) and complex dynamics (Meta-World, Half-Cheetah), and compare against several baselines including DPT, AD, and behavior cloning variants.

**Methods And Evaluation Criteria:**

Methodology:
- Weighted Maximum Likelihood Pretraining: DIT is trained using a weighted maximum likelihood objective where the weights are an exponential function of an estimated advantage function. This aims to give higher importance to actions that are deemed “better” in the historical data.
- In-Context Advantage Estimation: The paper introduces transformer-based modules to estimate the value and action-value functions in-context, thereby approximating the advantage for each state–action pair.
- Task-Conditioned Policy: The transformer model is conditioned on the context (a set of transitions from the environment) so that it can adapt its decision-making to new, unseen tasks during deployment.

Evaluation:
- The experiments are carried out in both online (where the agent gathers additional data) and offline (using fixed historical trajectories) settings.
- The evaluation metrics include cumulative return and regret (for bandit problems) and episode cumulative return (for MDPs), which are standard and appropriate for the problem domain.

**Other Comments Or Suggestions:**

- Too many sentences are marked with red, which hinders a smooth reading process. I would recommend to mark very few phrases that are truly important.
-  The Primary Area: Reinforcement Learning->Everything Else should be used sparsely. Is there a specific category that this paper falls into?

**Other Strengths And Weaknesses:**

Weaknesses
- According to the proposed method, it is extremely important to learn a universal advantage function that is robust to distribution shifts, which may not be scalable.
-  In context learning needs a few tokens to construct the context. What is the cold-start performance of ICRL? How to perform reliable zero-shot adaptation?
-  There seems to be a reload of meaning between the "In Context" RL and "In Context" Learning of LLMs.
- The theoretical results are less significant, considering that Prop. 4.1 is a well-known conclusion in RL, and the quadratic reliance on effective planning horizon in Prop. 4.2 makes the bound rather loose.

**Questions For Authors:**

See weaknesses.

**Relation To Broader Scientific Literature:**

N/A

**Theoretical Claims:**

I did not check the detailed derivations, but they seem reasonable as the conclusions are standard in RL literature.

---

> ### Author Rebuttal · Authors · 2025-04-01
>
> Thank you for your insightful comments. We hope our response can address your concerns.
>
> ### Presentation
>
> > **Style Improvement.** We will follow the reviewer’s suggestion to remove most of the colored text, keeping only minimal phrases with key importance.
>
> > **Clarification on the term “In-Context.”** We appreciate the reviewer highlighting this potential ambiguity. While “in-context learning” originated in the LLM literature, the term has recently been adopted by several works in reinforcement learning to describe similar ideas—namely, models that adapt to new tasks by conditioning on sequences of past transitions, without gradient updates. Our use of “in-context RL” follows this growing convention.
> Nonetheless, we agree that the dual usage can be confusing. In the revised manuscript, **we will explicitly distinguish in-context RL from ICL in language models to ensure clarity and avoid potential misunderstandings**.
>
> ### Methods.
>
> > **Zero-shot Adaptation.**  DIT builds on the supervised pretraining framework of DPT and can act as an online meta-policy in new environments. Specifically, **it can predict actions without conditioning on a context dataset, as the prediction is only based on the token corresponding to the query state**. In the zero-shot setting—i.e., with no context tokens—DIT and DPT default to the behavior learned during pretraining, effectively leveraging a strong prior. Cold-start performance is demonstrated in the online testing experiments (Figures 2, 3, and 4). These results show that **DIT quickly improves its performance in-context with more information (trajectories), adapting efficiently even in the cold-start setting**.
>
> ### Learned Advantage Function and Distribution Shift
>
> > We appreciate the reviewer’s insightful comment. We would like to clarify that **DIT does not require robustness to distribution shift during the pretraining**.
>
> > This is because the learned advantage functions are only applied to actions taken by the same behavioral policies that generated the trajectories—meaning **all computations remain in-distribution with respect to those behavioral policies**.
>
> > This design ensures stability and avoids the challenges typically associated with off-policy corrections. We consider this to be a key strength of DIT, and we will make this point more explicit in the revised manuscript.
>
> ### Clarification of Contributions
> > While similar theoretical results have been established in the standard RL setting, **our contribution lies in extending these insights to the in-context reinforcement learning (ICRL) framework**. Although theory is not the primary focus of this work, we include the analysis to better motivate and ground the design of DIT, **helping the reader understand why the method is effective**.
>
> > Our core insight is that if "relatively good" actions can be identified and emphasized during supervised pretraining, a transformer-based policy can match the performance of meta-policies trained with significantly more expensive data, such as DPT. Although individual weighted action labels may be noisy, our analysis and experiments show that **the weighted MLE objective—when averaged over diverse environments—can yield high-quality meta-policies capable of generalizing to unseen tasks**.
>
> > Furthermore, with the proposed in-context advantage estimator, we observe that transformer models can learn to generate reliable value function estimates across tasks and behavioral policies using only supervised learning. We believe these empirical insights are important to share, as they highlight the practicality and potential of ICRL.
>
> > On the practical side, **DIT is simple to implement and significantly improves the feasibility of in-context RL**, as suboptimal trajectories are much easier to collect in real-world systems, especially in industry settings where large amounts of historical data are already available. **This opens the door for broader adoption of ICRL methods in practical applications**.

---

### Decision · Program_Chairs · 2025-05-01

**Decision:**

Accept (poster)

**Comment:**

This paper proposes a method to learn effectively from suboptimal historical data without requiring optimal action labels for in-context reinforcement learning (ICRL). Reviewers generally appreciated the idea of using readily available suboptimal data for ICRL, which is more practical than methods requiring optimal labels (like DPT) or full learning histories (like AD). They also appreciated theoretical grounding and strong empirical results showing DIT performing competitively, and sometimes surprisingly better, than baselines like DPT trained under similar data constraints. There was a concern regarding why DPT performed lower than in its original paper and why DIT sometimes surpassed it, which was partially explained by the authors during the rebuttal period.

Overall, this paper presents a valuable and practical contribution to the ICRL field by enabling learning from more realistic, suboptimal datasets. During the post-rebuttal discussion period, the reviewers strongly suggested the authors to explain why DPT performs poorer than expected and make a more precise claim about DIT's performance relative to the "particular" implementation of DPT in the camera-ready version. Assuming that the authors will make this change, I recommend to accept this paper.